# SF(DA)²: SOURCE-FREE DOMAIN ADAPTATION THROUGH THE LENS OF DATA AUGMENTATION

**Uiwon Hwang**[1]  **Jonghyun Lee**[2]  **Juhyeon Shin**[3]  **Sungroh Yoon**[2,3,*]

[1] Division of Digital Healthcare, Yonsei University
[2] Department of Electrical and Computer Engineering, Seoul National University
[3] Interdisciplinary Program in Artificial Intelligence, Seoul National University
uiwon.hwang@yonsei.ac.kr, {leejh9611, newjh12, sryoon}@snu.ac.kr

## ABSTRACT

In the face of the deep learning model's vulnerability to domain shift, source-free domain adaptation (SFDA) methods have been proposed to adapt models to new, unseen target domains without requiring access to source domain data. Although the potential benefits of applying data augmentation to SFDA are attractive, several challenges arise such as the dependence on prior knowledge of class-preserving transformations and the increase in memory and computational requirements. In this paper, we propose Source-free Domain Adaptation Through the Lens of Data Augmentation (SF(DA)²), a novel approach that leverages the benefits of data augmentation without suffering from these challenges. We construct an augmentation graph in the feature space of the pretrained model using the neighbor relationships between target features and propose spectral neighborhood clustering to identify partitions in the prediction space. Furthermore, we propose implicit feature augmentation and feature disentanglement as regularization loss functions that effectively utilize class semantic information within the feature space. These regularizers simulate the inclusion of an unlimited number of augmented target features into the augmentation graph while minimizing computational and memory demands. Our method shows superior adaptation performance in SFDA scenarios, including 2D image and 3D point cloud datasets and a highly imbalanced dataset.

## 1 INTRODUCTION

In recent years, deep learning has achieved significant advancements and is widely explored for real-world applications. However, the performance of deep learning models can significantly deteriorate when deployed on unlabeled target domains, which differ from the source domain where the training data was collected. This *domain shift* poses a challenge for applying deep learning models in practical scenarios. To address this, various domain adaptation (DA) methods have been proposed to adapt the model to new, unseen target domains (Chen et al., 2019; Xu et al., 2019; Jin et al., 2020; Na et al., 2021; Long et al., 2013; Ganin & Lempitsky, 2015; Tzeng et al., 2017; Qin et al., 2019). Traditional domain adaptation techniques require both source domain and target domain data. However, practical limitations arise when source domain data is inaccessible or difficult to obtain due to cost or privacy concerns. Source-free domain adaptation (SFDA) overcomes the need for direct access to the source domain data by using only a model pretrained on the source domain data. SFDA methods focus on adapting the model to the target domain via unsupervised or self-supervised learning, which leverages the class semantic information learned from the source domain data (Liang et al., 2020; Yang et al., 2021a; Lee et al., 2022; Yang et al., 2022; Zhang et al., 2022).

Recent studies (Chen et al., 2022; Zhang et al., 2022) have explored using data augmentation to enhance adaptation performance by enriching target domain information with transformations such as flipping or rotating images. These methods, however, require prior knowledge to ensure the preservation of class semantic information within augmented data. For instance, applying a 180-degree rotation to an image of the digit 6 would result in the digit 9, which compromises the class semantic information. Moreover, increasing the number of augmented samples can lead to higher

---

*Corresponding author

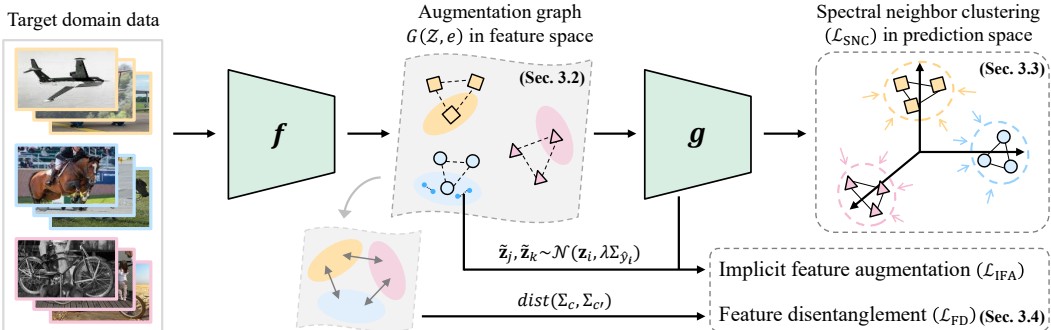

Figure 1: Overview of SF(DA)$^2$. Here, $f$ and $g$ indicate the feature extractor and the classifier, $\mathbf{z}$ denotes a target feature, $\Sigma$ is a covariance matrix for a class, and $dist(\cdot, \cdot)$ denotes a distance measure.

memory usage and computational load. In response to these challenges, we aim to devise an SFDA method that effectively harnesses the advantages of data augmentation while mitigating its drawbacks.

In this paper, we propose a novel approach named Source-free Domain Adaptation Through the Lens of Data Augmentation (SF(DA)$^2$). We present a unique perspective of SFDA inspired by data augmentation, which leads us to the introduction of an augmentation graph in the feature space of the pretrained model. The augmentation graph depicts relationships among target features using class semantic information learned by the pretrained model. Building upon the augmentation graph, the proposed method consists of spectral neighborhood clustering (SNC), designed for maximal utilization of the information from target domain data samples, and implicit feature augmentation (IFA) along with feature disentanglement (FD), designed to leverage additional information from the estimated distribution of the target domain data.

More specifically, we propose SNC within the prediction space to partition the augmentation graph via spectral clustering. The SNC loss promotes clustering in the prediction space and guides the classifier in assigning clusters, which facilitates effective adaptation. Furthermore, we explore the inclusion of augmented target features (vertices) into the augmentation graph. This allows us to identify better clusters, which might be challenging to discover using only the given target domain data samples. To this end, we propose two regularization strategies: IFA and FD. We derive the IFA loss as a closed-form upper bound for the expected loss over an infinite number of augmented features. This formulation of IFA minimizes computational and memory overhead. The FD loss serves as a crucial component in preserving class semantic information within the augmented features by disentangling the feature space.

We performed experiments on challenging benchmark datasets, including VisDA (Peng et al., 2017), DomainNet (Peng et al., 2019), PointDA (Qin et al., 2019), and VisDA-RSUT (Li et al., 2021). We verified that our method outperforms existing state-of-the-art methods on 2D image, 3D point cloud, and highly imbalanced datasets. Furthermore, we observed that IFA and FD effectively boost the performances of existing methods. The contributions of this work can be summarized as follows:

- We provide a fresh perspective on SFDA by interpreting it through the lens of data augmentation. Then, we propose SF(DA)$^2$ that thoroughly harnesses intuitions derived from data augmentation without explicit augmentation of target domain data.

- We propose the spectral neighborhood clustering (SNC) loss for identifying partitions in the augmentation graph via spectral clustering. This approach effectively clusters the target domain data in the prediction space.

- We derive the implicit feature augmentation (IFA) loss, which enables us to simulate the effects of an unlimited number of augmented features while maintaining minimal computational and memory overhead. To supplement the effectiveness of IFA, we propose the feature disentanglement (FD) loss. This further regularizes the feature space to learn distinct class semantic information along different directions.

- We validate the effectiveness of the proposed method through experiments performed on challenging 2D image and 3D point cloud datasets. Our method significantly outperforms existing methods in the SFDA settings.

## 2 RELATED WORK

**Source-free Domain Adaptation**    SHOT (Liang et al., 2020) utilizes information maximization and pseudo-labeling with frozen classifier weights. CoWA-JMDS (Lee et al., 2022) estimates a Gaussian mixture in the feature space to obtain target domain information. NRC (Yang et al., 2021a) and AaD (Yang et al., 2022) are rooted in local consistency between neighbors in the feature space. Contrastive learning-based methods such as DaC (Zhang et al., 2022) and AdaContrast (Chen et al., 2022) involve data augmentation, which requires not only prior knowledge to preserve class semantic information within augmented data but also computational and memory overhead. Our method circumvents these issues, providing efficient adaptation without explicit data augmentation.

**Data Augmentation**    Data augmentation improves the generalization performance of the model by applying class-preserving transformations to training data and incorporating the augmented data in model training. To minimize the reliance on prior knowledge, several studies propose optimization techniques to find combinations of transformations within constrained search spaces (Cubuk et al., 2019; Lee et al., 2020; Zheng et al., 2022). ISDA (Wang et al., 2019) implicitly augments data by optimizing an upper bound of the expected cross-entropy loss of augmented features. We propose implicit feature augmentation tailored for SFDA settings.

**Spectral Contrastive Learning**    HaoChen et al. (2021) propose contrastive loss with the augmentation graph, connecting augmentations of the same data sample and performing spectral decomposition. They prove that minimizing this loss achieves linearly separable features. Our work is motivated by spectral contrastive learning but with two key differences. First, we define the augmentation graph in the feature space, leveraging class semantic information learned by the pretrained model to directly enhance predictive performance on target domain data. Second, our method does not rely on explicit data augmentation for positive pairs; instead, we utilize neighbors in the feature space as positive pairs, considering them as highly nonlinear transformation relationships. We further discuss differences between existing work and the proposed method, as well as additional related work in Appendix D.

## 3 PROPOSED METHOD

In the following sections, we propose Source-free Domain Adaptation Through the Lens of Data Augmentation (SF(DA)$^2$). We first formulate the scenario of SFDA (Section 3.1). From the intuitions of data augmentation, we define the augmentation graph in the feature space of the pretrained model (Section 3.2), find partitions of the augmentation graph (Section 3.3), and implicitly augment features (Section 3.4). An overview of SF(DA)$^2$ is illustrated in Figure 1.

### 3.1 SOURCE-FREE DOMAIN ADAPTATION SCENARIO

We consider source domain data $\mathcal{D}_s = \{(\mathbf{x}_i^s, y_i^s)\}_{i=1}^{M_s}$, where the class labels $y_i^s$ belong to $C$ classes. We also consider unlabeled target domain data $\mathcal{D}_t = \{\mathbf{x}_i^t\}_{i=1}^{M_t}$ sampled from target domain data distribution $P(\mathcal{X}^t)$. Target domain data have the same $C$ classes as $\mathcal{D}_s$ in this paper (closed-set setting). In SFDA scenarios, we have access to a source pretrained model consisting of a feature extractor $f$ and a classifier $g$. The feature extractor $f$ takes a target domain data sample as input and generates target features $\mathbf{z}_i = f(\mathbf{x}_i^t)$. The classifier $g$ consists of a fully connected layer and predicts classes from the target features $p_i = \delta(g(\mathbf{z}_i))$ where $\delta$ denotes the softmax function.

### 3.2 AUGMENTATION GRAPH ON FEATURE SPACE

In the proposed method, we focus on the relationships between target domain data samples within the feature space of the pretrained model. We then build an augmentation graph grounded in our intuitions on the SFDA scenario.

Deep neural networks trained on source domain data encode class semantic information into distinct directions in their feature space (Wang et al., 2019; Bengio et al., 2013). For example, different hairstyles for the 'person' class or wing shapes for the 'airplane' class can be represented along specific directions. This property enables the pretrained model to map target domain data samples

with similar class semantic information close to one another. From this motivation, we can make the following assumption:

**Intuition 1** (Clustering assumption of source model). *Target domain data that share the same semantic information are mapped to their **neighbors** in the feature space of the pretrained model. In this context, neighbors refer to target features with small cosine distances to a given target feature.*

We also hypothesize that target domain data samples with shared class semantic information can be connected through highly nonlinear functions, representing potential augmentation relationships.

**Intuition 2** (Augmentation assumption of target domain data). *Target domain data sharing class semantic information may have highly nonlinear functions to **transform each other**.*

To formalize our intuitions of the relationships between target domain data, we introduce the concept of an augmentation graph (HaoChen et al., 2021). The augmentation graph consists of vertices representing target domain data and edge weights representing the probability of augmentation relationships between the target domain data. From our intuitions, samples with shared semantic information are neighbors in the feature space. These samples have also augmentation relationships, enabling their connection in the augmentation graph. Consequently, we can construct the augmentation graph on the feature space, which provides a structured representation of target domain data.

Given a set of all target features in the *population* distribution of the target domain $\mathcal{Z} = \{\mathbf{z} = f(\mathbf{x}^t)|\mathbf{x}^t \sim P(\mathcal{X}^t)\}$, we define the population augmentation graph $G(\mathcal{Z}, e)$ with edges $e_{ij}$ between two target features $\mathbf{z}_i$ and $\mathbf{z}_j$ as follows:

$$e_{ij} = e\left(\mathbf{z}_i, \mathbf{z}_j\right) = Pr(\mathbf{z}_j \in N_i) \tag{1}$$

where $N_i$ is the set of neighbors of $\mathbf{z}_i$.

### 3.3 Finding Partition on Prediction Space

Using the augmentation graph, our next objective is to find meaningful partitions or clusters of the graph that represent different classes. To achieve this, we employ spectral clustering on the graph (Shi & Malik, 2000; Ng et al., 2001), which reveals the underlying structure of the data. In particular, HaoChen et al. (2021) proposed a loss function that performs spectral clustering on the augmentation graph:

**Lemma 1** (HaoChen et al. (2021)). *Let $L$ be the normalized Laplacian matrix of the augmentation graph $G(\mathcal{X}, l)$. The matrix $H$, which has the eigenvectors corresponding to the $k$ largest eigenvalues of $L$ as its columns, can be learned as a function $h$ by minimizing the following matrix factorization loss:*

$$\min_h \mathcal{L}(h) = \|(I - L) - HH^T\|_F^2 \tag{2}$$

$$= \mathrm{const} - 2\sum_{i,j} \frac{l_{x_i x_j}}{\sqrt{l_{x_i}}\sqrt{l_{x_j}}} h(x_i)^T h(x_j) + \sum_{i,j} \left(h(x_i)^T h(x_j)\right)^2 \tag{3}$$

*where $x_i$ and $x_j$ are vertices of the augmentation graph, $l_{x_i x_j}$ is an edge between $x_i$ and $x_j$, $l_{x_i} = \sum_j l_{x_i x_j}$ denotes the degree of a vertex $x_i$, and $\mathrm{const}$ is a constant term.*

Using Lemma 1, we describe our approach for finding partitions in the augmentation graph, focusing on the prediction space. Since we have the target domain data $\mathcal{D}_t$ sampled from $P(\mathcal{X}^t)$, we now build an instance of the population augmentation graph denoted by $\hat{G}$. Each target feature $\mathbf{z}_i$ is represented as a vertex. We consider edges only between the $K$-nearest neighbors for each vertex, assuming that these neighbors share the same class semantic information. The edge weight $e_{ij}$ is set to 1 if $\mathbf{z}_j$ is among the $K$-nearest neighbors of $\mathbf{z}_i$ in the feature space, denoted by $N_i^K$. To efficiently retrieve the nearest neighbors, we utilize two memory banks $\mathcal{F}$ and $\mathcal{S}$ to retain all target features and their predictions as used in the previous studies (Yang et al., 2022; 2021a; Saito et al., 2020). Since all samples are i.i.d. and each target feature has the same number of neighbors, the degree of all target features is equal (i.e., $e_i = e_j = K$ for $\forall i, j \in [0, M_t]$). Given a mini-batch $B = \{\mathbf{x}_i^t\}_{i=1}^b$ of target domain data samples, we can derive the spectral neighborhood clustering (SNC) loss on $\hat{G}$:

$$\mathcal{L}_{\mathrm{SNC}}(p_i) = -\frac{2}{K}\sum_{j \in N_i^K} p_i^T p_j + \sum_{k \in B} \left(p_i^T p_k\right)^2 \tag{4}$$

The first term attracts the predictions of neighborhood features, leading to tight clusters in the prediction space, similar to previous approaches (Yang et al., 2022; 2021a;b). In contrast, the second term encourages distinct separation between different clusters by driving the predictions of all target features apart from each other. We further incorporate a decay factor into the second term to make predictions more certain as adaptation proceeds. As suggested in (Yang et al., 2022), we multiply the second term by $(1 + 10 \times \frac{\text{iter}}{\text{max\_iter}})^{-\beta}$ where we set $\beta$ to 5 in our experiments. Overall, the SNC loss facilitates clustering in the prediction space and supports cluster assignment by the classifier $g$.

### 3.4 IMPLICIT FEATURE AUGMENTATION

In this section, we aim to obtain an improved partition by further approximating the population augmentation graph $G$. We achieve this by augmenting target features connected to individual target features. The key focus of this section is to perform feature augmentation without relying on prior knowledge.

To preserve class semantic information while augmenting target features, we revisit the property of deep learning models (Wang et al., 2019; Bengio et al., 2013), which suggests that class semantic information is represented by specific directions and magnitudes in the feature space. We discern the direction and magnitude for each class by estimating the class-wise covariance matrix of target features (denoted as $\mathbf{\Sigma} = \{\Sigma_c\}_{c=1}^{C}$) online during adaptation using mini-batch statistics. The detailed method for online covariance estimation, adopted from ISDA (Wang et al., 2019), is described in Appendix C.3. As target features lack class labels, we use their pseudo-labels $\hat{y}_i = \arg\max_c p_i$ to update the covariance matrices accordingly.

Using the estimated covariance matrices, we can preserve class semantic information in augmented features by sampling features from a Gaussian distribution. Since an edge (Equation 1) connects a target feature with its *neighbors*, we set the mean of the Gaussian distribution to the target feature itself, and we use the estimated covariance matrix corresponding to the pseudo-label as the variance:

$$\tilde{\mathbf{z}}_j, \tilde{\mathbf{z}}_k, \cdots \sim \mathcal{N}(\mathbf{z}_i, \lambda\Sigma_{\hat{y}_i}) \tag{5}$$

At the beginning of adaptation, the estimation of covariance matrices may be inaccurate. Therefore, following Wang et al. (2019), we set $\lambda = \lambda_0 \times \frac{\text{iter}}{\text{max\_iter}}$ and $\lambda_0 = 5$ in experiments. This gradually increases the impact of the estimated covariance matrices on the model as the adaptation proceeds.

Augmented features can be used in calculating the first term of the SNC loss to promote similar predictions. However, directly sampling features from the Gaussian distribution during adaptation can increase training time and memory usage, undesirable in the SFDA context. Instead, we aim to derive an upper bound for the expected SNC loss. Since obtaining a closed-form upper bound for the first term of the SNC loss is intractable, we consider its logarithm. This simplifies the solution while preserving the goal of adaptation. The explicit feature augmentation (EFA) loss of two augmented features, $\tilde{\mathbf{z}}_j$ and $\tilde{\mathbf{z}}_k$, sampled from the estimated neighbor distribution of $\mathbf{z}_i$, is expressed as follows:

$$\mathcal{L}_{\text{EFA}}(\tilde{\mathbf{z}}_j, \tilde{\mathbf{z}}_k) = -\log \tilde{p}_j^T \tilde{p}_k \tag{6}$$

where $\tilde{p}_j = \delta(g(\tilde{\mathbf{z}}_j))$ and $\tilde{p}_k = \delta(g(\tilde{\mathbf{z}}_k))$ are the predictions of the augmented features.

We then derive an upper bound for the expected EFA loss and propose the implicit feature augmentation (IFA) loss as follows:

**Proposition 1.** *Suppose that $\tilde{\mathbf{z}}_j, \tilde{\mathbf{z}}_k \sim \mathcal{N}(\mathbf{z}_i, \lambda\Sigma_{\hat{y}_i})$, then we have an upper bound of expected $\mathcal{L}_{\text{EFA}}$ for an infinite number of augmented features, which we call implicit feature augmentation loss:*

$$\mathcal{L}_{\text{EFA}}^{\infty}(\mathbf{z}_i; f, g) = \mathbb{E}_{\tilde{\mathbf{z}}_j \sim \mathcal{N}(\mathbf{z}_i, \lambda\Sigma_{\hat{y}_i})} \left[ \mathbb{E}_{\tilde{\mathbf{z}}_k \sim \mathcal{N}(\mathbf{z}_i, \lambda\Sigma_{\hat{y}_i})} \left[ -\log \tilde{p}_j^T \tilde{p}_k \right] \right] \tag{7}$$

$$\leq -2\sum_{c=1}^{C} \log \frac{\exp(g(\mathbf{z}_i)_c)}{\sum_{c'=1}^{C} \exp\left(g(\mathbf{z}_i)_{c'} + \frac{\lambda}{2}(w_{c'} - w_c)^T \Sigma_{\hat{y}_i}(w_{c'} - w_c)\right)} = \mathcal{L}_{\text{IFA}}(\mathbf{z}_i, \Sigma_{\hat{y}_i}, g) \tag{8}$$

*where $g(\cdot)_c$ denotes the classifier output (logit) for the $c$-th class, and $w_c$ is the weight vector for the $c$-th class of the classifier $g$.*

Proof of Proposition 2 is provided in Appendix B. Optimizing the IFA loss allows us to achieve the effect of sampling an infinite number of features with minimal extra computation and memory usage.

In the denominator of the IFA loss, the first term is proportional to the prediction for the $c'$-th class of the target feature. The second term relates to the square of the cosine similarity between the normal vector of the decision boundary separating the $c$-th and $c'$-th classes, and the direction of feature augmentation. Namely, the denominator can be interpreted as a regularization term that aligns the decision boundary between the predicted and other classes with the principal direction of the covariance matrix of the predicted class. This is in accordance with the findings by Balestriero et al. (2022), which suggest that the explicit regularizer of data augmentation encourages the kernel space of the model's Jacobian matrix to be aligned with the principal direction of the tangent space of the augmented data manifold. Recalling that the target feature is augmented to the direction of variance of target features with the same pseudo-label, the IFA loss promotes the decision boundaries for each class to align with the principal direction of the variance of target features with the same pseudo-label.

Maintaining class semantics in augmented features relies heavily on a well-disentangled feature space. If similar classes, such as cars and trucks, have similar covariances, it suggests that the feature space may not capture their semantic differences sufficiently. This could lead to non-class-preserving transformations during augmentation, possibly undermining the model's performance when the IFA loss is incorporated. To mitigate this, a more disentangled feature space is needed, ensuring a distinct representation for each class.

To encourage each direction in the feature space to represent different semantics (i.e., to promote a disentangled space), we maximize the cosine distance between covariance matrices corresponding to similar classes. This indicates that features with different pseudo-labels are distributed in distinct directions. Consequently, the semantic information of different classes is encoded in separate directions within the feature space, resulting in a disentangled feature space. We propose the feature disentanglement (FD) loss as follows:

$$\mathcal{L}_{\text{FD}} = -\frac{1}{2} \sum_{i,j} a_{ij} \left( 1 - \frac{\text{tr}\{\Sigma_i \, \Sigma_j\}}{\|\Sigma_i\|_F \|\Sigma_j\|_F} \right) \tag{9}$$

where $\Sigma_i$ and $\Sigma_j$ denote the covariance matrices for the $i$-th and $j$-th classes, respectively, and the weight $a_{ij}$ serves as a measure of similarity between $i$-th and $j$-th classes, and it is calculated as the dot product of the mean prediction vector (i.e., $a_{ij} = \bar{p}_i^T \bar{p}_j$, where $\bar{p}_c = \frac{1}{|\{i:\hat{y}_i=c\}|} \sum_{i \in \{i:\hat{y}_i=c\}} p_i$). Therefore, the FD loss places greater emphasis on class pairs that exhibit similar predictions on the target domain data. The weights are calculated at the beginning of each epoch.

The final objective for adaptation can be expressed as follows:

$$\min_{f,g} \mathcal{L}_{\text{SNC}} + \alpha_1 \mathcal{L}_{\text{IFA}} + \alpha_2 \mathcal{L}_{\text{FD}} \tag{10}$$

where $\alpha_1$ and $\alpha_2$ denote hyperparameters. The procedure of SF(DA)$^2$ is presented in Algorithm 1.

## 4 EXPERIMENTS

In this section, we evaluate the performance of SF(DA)$^2$ on several benchmark datasets: Office-31 (Saenko et al., 2010), VisDA (Peng et al., 2017), DomainNet (Peng et al., 2019), PointDA-10 (Qin et al., 2019), and VisDA-RSUT (Li et al., 2021). The results on Office-31 and details of the datasets are provided in Appendix A.1 and C.2, respectively.

**Implementation details**  For a fair comparison, we adopt identical network architectures, optimizers, and batch sizes as benchmark methods (Liang et al., 2020; Yang et al., 2021a; 2022).

---

**Algorithm 1** Adaptation procedure of SF(DA)$^2$

**Require:** $f$ and $g$ (trained on $\mathcal{D}_s$), $\mathcal{D}_t = \{\mathbf{x}_i^t\}_{i=1}^{M_t}$
1: **while** training loss is not converged **do**
2:     **if** epoch start **then**
3:         Update $a_{ij}$ for FD loss
4:     **end if**
5:     Sample batch $B$ from $\mathcal{D}_t$ and update $\mathcal{F}, \mathcal{S}$
6:     Retrieve neighbors $\mathcal{N}_i^K$ for each $\mathbf{z}_i$ in $B$
7:     Update $f$ and $g$ using SGD
8:         $\nabla_{f,g} \, \mathcal{L}_{\text{SNC}} + \alpha_1 \mathcal{L}_{\text{IFA}} + \alpha_2 \mathcal{L}_{\text{FD}}$
9: **end while**

---

We run our methods with three different random seeds and report the average accuracies. **SF** in the tables denotes source-free. More implementation details are presented in Appendix C.1.

Most hyperparameters of our method do not require heavy tuning. We set $K$ to 5 on VisDA, PointDA-10, and VisDA-RSUT, and 2 on DomainNet. We set $\alpha_1$ to 1e-4 on VisDA, DomainNet, and

Table 1: Accuracy (%) on the VisDA dataset (ResNet-101).

| Method | SF | plane | bicycle | bus | car | horse | knife | mcycl | person | plant | sktbrd | train | truck | Per-class |
|---|---|---|---|---|---|---|---|---|---|---|---|---|---|---|
| BSP (Chen et al., 2019) | ✗ | 92.4 | 61.0 | 81.0 | 57.5 | 89.0 | 80.6 | 90.1 | 77.0 | 84.2 | 77.9 | 82.1 | 38.4 | 75.9 |
| SAFN (Xu et al., 2019) | ✗ | 93.6 | 61.3 | 84.1 | 70.6 | 94.1 | 79.0 | 91.8 | 79.6 | 89.9 | 55.6 | 89.0 | 24.4 | 76.1 |
| MCC (Jin et al., 2020) | ✗ | 88.7 | 80.3 | 80.5 | 71.5 | 90.1 | 93.2 | 85.0 | 71.6 | 89.4 | 73.8 | 85.0 | 36.9 | 78.8 |
| FixBi (Na et al., 2021) | ✗ | 96.1 | 87.8 | 90.5 | 90.3 | 96.8 | 95.3 | 92.8 | 88.7 | 97.2 | 94.2 | 90.9 | 25.7 | 87.2 |
| Source only (He et al., 2016) | - | 60.9 | 21.6 | 50.9 | 67.6 | 65.6 | 6.3 | 82.2 | 23.2 | 57.3 | 30.6 | 84.6 | 8.0 | 46.6 |
| 3C-GAN (Li et al., 2020) | ✓ | 94.8 | 73.4 | 68.8 | 74.8 | 93.1 | 95.4 | 88.6 | 84.7 | 89.1 | 84.7 | 83.5 | 48.1 | 81.6 |
| SHOT (Liang et al., 2020) | ✓ | 94.6 | 87.5 | 80.4 | 59.5 | 92.9 | 95.1 | 83.1 | 80.2 | 90.9 | 89.2 | 85.8 | 56.9 | 83.0 |
| NRC (Yang et al., 2021a) | ✓ | 96.1 | 90.8 | 83.9 | 61.5 | 95.7 | 95.7 | 84.4 | 80.7 | 94.0 | 91.9 | 89.0 | 59.5 | 85.3 |
| CoWA-JMDS (Lee et al., 2022) | ✓ | 96.2 | 90.6 | 84.2 | 75.5 | 96.5 | 97.1 | 88.2 | 85.6 | 94.9 | 93.0 | 89.2 | 53.5 | 87.0 |
| AaD (Yang et al., 2022) | ✓ | 96.8 | 89.3 | 83.8 | 82.8 | 96.5 | 95.2 | 90.0 | 81.0 | 95.7 | 92.9 | 88.9 | 54.6 | 87.3 |
| **SF(DA)$^2$** | ✓ | 96.8 | 89.3 | 82.9 | 81.4 | 96.8 | 95.7 | 90.4 | 81.3 | 95.5 | 93.7 | 88.5 | 64.7 | **88.1** |

Table 2: Accuracy (%) on 7 domain shifts of the DomainNet-126 dataset (ResNet-50).

| Method | SF | S→P | C→S | P→C | P→R | R→S | R→C | R→P | Avg. |
|---|---|---|---|---|---|---|---|---|---|
| MCC (Jin et al., 2020) | ✗ | 47.3 | 34.9 | 41.9 | 72.4 | 35.3 | 44.8 | 65.7 | 48.9 |
| Source only (He et al., 2016) | - | 50.1 | 46.9 | 53.0 | 75.0 | 46.3 | 55.5 | 62.7 | 55.6 |
| TENT (Wang et al., 2021) | ✓ | 52.4 | 48.5 | 57.9 | 67.0 | 54.0 | 58.5 | 65.7 | 57.7 |
| SHOT (Liang et al., 2020) | ✓ | 66.1 | 60.1 | 66.9 | 80.8 | 59.9 | 67.7 | 68.4 | 67.1 |
| NRC (Yang et al., 2021a) | ✓ | 65.7 | 58.6 | 64.5 | 82.3 | 58.4 | 65.2 | 68.2 | 66.1 |
| AaD (Yang et al., 2022) | ✓ | 65.4 | 54.2 | 59.8 | 81.8 | 54.6 | 60.3 | 68.5 | 63.5 |
| **SF(DA)$^2$** | ✓ | 67.7 | 59.6 | 67.8 | 83.5 | 60.2 | 68.8 | 70.5 | **68.3** |

Table 3: Comparison of SF(DA)$^2$++ and other two-stage methods on VisDA (ResNet-101).

| Method | SHOT++ (Liang et al., 2021) | feat-mixup + SHOT++ (Kundu et al., 2022) | **SF(DA)$^2$++** |
|---|---|---|---|
| Per-class | 87.3 | 87.8 | **89.6** |

Table 4: Accuracy (%) on the PointDA-10 dataset (PointNet).

| Method | SF | Model→Shape | Model→Scan | Shape→Model | Shape→Scan | Scan→Model | Scan→Shape | Avg. |
|---|---|---|---|---|---|---|---|---|
| MMD (Long et al., 2013) | ✗ | 57.5 | 27.9 | 40.7 | 26.7 | 47.3 | 54.8 | 42.5 |
| DANN (Ganin & Lempitsky, 2015) | ✗ | 58.7 | 29.4 | 42.3 | 30.5 | 48.1 | 56.7 | 44.2 |
| ADDA (Tzeng et al., 2017) | ✗ | 61.0 | 30.5 | 40.4 | 29.3 | 48.9 | 51.1 | 43.5 |
| MCD (Saito et al., 2018) | ✗ | 62.0 | 31.0 | 41.4 | 31.3 | 46.8 | 59.3 | 45.3 |
| PointDAN (Qin et al., 2019) | ✗ | 64.2 | 33.0 | 47.6 | 33.9 | 49.1 | 64.1 | 48.7 |
| Source only (Qi et al., 2017) | - | 43.1 | 17.3 | 40.0 | 15.0 | 33.9 | 47.1 | 32.7 |
| VDM-DA (Tian et al., 2021) | ✓ | 58.4 | 30.9 | 61.0 | 40.8 | 45.3 | 61.8 | 49.7 |
| NRC (Yang et al., 2021a) | ✓ | 64.8 | 25.8 | 59.8 | 26.9 | 70.1 | 68.1 | 52.6 |
| AaD (Yang et al., 2022) | ✓ | 69.6 | 34.6 | 67.7 | 28.8 | 68.0 | 66.6 | 55.9 |
| **SF(DA)$^2$** | ✓ | 70.3 | 35.5 | 68.3 | 29.0 | 70.4 | 69.2 | **57.1** |

PointDA-10, and 1e-3 on VisDA-RSUT. We set $\alpha_2$ to 10 on VisDA, PointNet-10, and VisDA-RSUT, and 1 on DomainNet.

## 4.1 EVALUATION RESULTS

**2D and 3D Datasets** We compare SF(DA)$^2$ with the source-present and source-free DA methods on 2D image datasets. For VisDA, we reproduce SHOT, NRC, CoWA-JMDS, AaD using their official codes and the pretrained models released by SHOT (Liang et al., 2020). As shown in Table 1, our method outperforms all other methods on VisDA in terms of average accuracy. We find similar observations on the results on DomainNet in Table 2.

Additionally, we extend SF(DA)$^2$ into a two-stage version called SF(DA)$^2$++. To ensure a fair comparison, we utilize the second training stage, which follows the same approach as SHOT++ (Liang et al., 2021). The results presented in Table 3 demonstrate that SF(DA)$^2$++ outperforms other two-stage methods on the VisDA dataset.

We also conduct comparisons on a 3D point cloud dataset, PointDA-10. In Table 4, our method significantly outperforms not only PointDAN (Qin et al., 2019) especially designed for *source-present* domain adaptation on point cloud data, but also source-free methods by a large margin (about 4.5%p). These results clearly demonstrate the effectiveness of SF(DA)$^2$ in domain adaptation.

Table 5: Accuracy (%) on the VisDA-RSUT dataset (ResNet-101).

| Method | SF | plane | bicycle | bus | car | horse | knife | mcycl | person | plant | sktbrd | train | truck | Per-class |
|---|---|---|---|---|---|---|---|---|---|---|---|---|---|---|
| DANN (Ganin & Lempitsky, 2015) | ✗ | 71.7 | 35.7 | 58.5 | 21.0 | 80.9 | 73.0 | 45.7 | 23.7 | 12.2 | 4.3 | 1.5 | 0.9 | 35.8 |
| BSP (Chen et al., 2019) | ✗ | 100.0 | 57.1 | 68.9 | 56.8 | 83.7 | 26.7 | 78.7 | 16.2 | 63.7 | 1.9 | 0.1 | 0.1 | 46.2 |
| MCD (Saito et al., 2018) | ✗ | 63.0 | 41.4 | 84.0 | 67.3 | 86.6 | 93.9 | 85.6 | 76.3 | 84.1 | 11.3 | 5.0 | 3.0 | 58.5 |
| Source only (He et al., 2016) | - | 79.7 | 15.7 | 40.6 | 77.2 | 66.8 | 11.1 | 85.1 | 12.9 | 48.3 | 14.3 | 64.6 | 3.3 | 43.3 |
| SHOT (Liang et al., 2020) | ✓ | 86.2 | 48.1 | 77.0 | 62.8 | 92.0 | 66.2 | 90.7 | 61.3 | 76.9 | 73.5 | 67.2 | 9.1 | 67.6 |
| CoWA-JMDS (Lee et al., 2022) | ✓ | 63.8 | 32.9 | 69.5 | 59.9 | 93.2 | 95.4 | 92.3 | 69.4 | 85.1 | 68.4 | 64.9 | 32.3 | 68.9 |
| NRC (Yang et al., 2021a) | ✓ | 86.2 | 47.6 | 66.7 | 68.1 | 94.7 | 76.6 | 93.7 | 63.6 | 87.3 | 89.0 | 83.6 | 20.5 | 73.1 |
| AaD (Yang et al., 2022) | ✓ | 73.9 | 33.3 | 56.6 | 71.4 | 90.1 | 97.0 | 91.9 | 70.8 | 88.1 | 87.2 | 81.2 | 39.4 | 73.4 |
| **SF(DA)**[2] | ✓ | 79.0 | 43.3 | 73.6 | 74.7 | 92.8 | 98.3 | 93.4 | 79.1 | 90.1 | 87.5 | 81.1 | 34.2 | **77.3** |

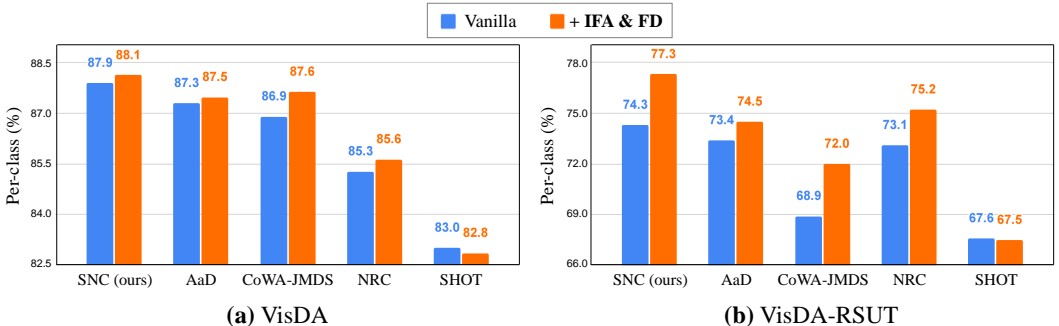

Figure 2: Effectiveness of implicit feature augmentation.

**Imbalanced Dataset**   In real-world domain adaptation scenarios, the classes of data can often be imbalanced. In order to evaluate our method in these scenarios, we perform a comparison of our method with competitive benchmarks on VisDA-RSUT, a dataset with highly imbalanced classes. For VisDA-RSUT, we reproduce SHOT, NRC, CoWA-JMDS, AaD using their official codes. The results presented in Table 5 demonstrate that our method surpasses the baseline methods by a significant margin (about 4%p). The results show the robustness of our method on extreme class imbalance. We also present results using additional metrics in Appendix A.2

## 4.2 ANALYSIS

**Effectiveness of Implicit Feature Augmentation**   Figure 2 illustrates the performance improvement of existing SFDA methods and our SNC after incorporating IFA and FD losses. The results highlight that IFA and FD improve most existing SFDA methods. Remarkably, their effectiveness is prominent in VisDA-RSUT, as shown in Figure 2 (b), leading to a 3%p performance gain on CoWA-JMDS and SNC. The significant improvement from incorporating IFA and FD into SNC can be attributed to their effective feature augmentation on the augmentation graph. The substantial performance gain in CoWA-JMDS can stem from its use of Gaussian distribution estimation in the feature space. For AaD and NRC, their use of neighbor information in the feature space aligns well with the assumptions of IFA, leading to performance enhancement. However, the assumptions of SHOT do not align well with IFA, resulting in a slight performance decrease when IFA loss is incorporated.

**Loss Functions for Implicit Feature Augmentation**   We conduct ablation studies on the VisDA-RSUT dataset to understand the impact of IFA and FD loss functions. In Figure 3, we measure (a) average accuracy and (b) the cosine similarity between the weight vector $w$ and the largest eigenvector $v$ of the estimated covariance matrix of the corresponding class. During adaptation with SNC, we observe a consistent increment in the cosine similarity.

We then analyze the impact of adding IFA or FD loss alone to SNC loss. Firstly, adding IFA alone decreases the cosine similarity, aligning the Jacobian matrix of the classifier ($w$) with the principal direction of the tangent space of the augmented data manifold (orthogonal with $v$), as discussed in Section 3.4. However, the sole addition of IFA results in a performance decrease. If different directions in the feature space do not clearly correspond to distinct class semantics, performing feature augmentation in such an entangled feature space may not preserve class information, causing

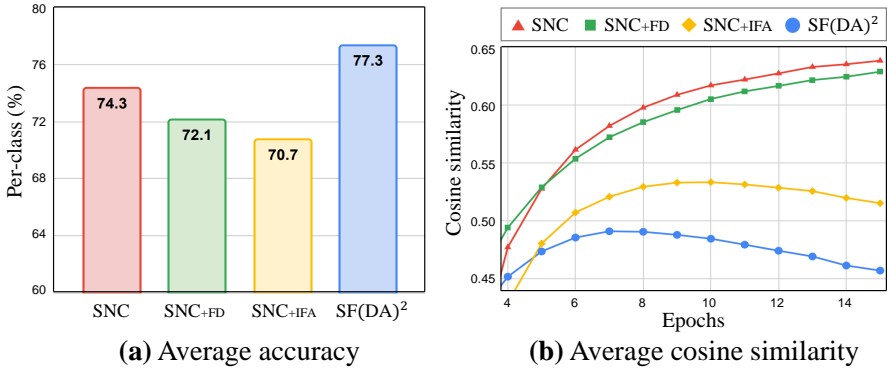

(a) Average accuracy        (b) Average cosine similarity

Figure 3: Ablation study on loss functions for implicit feature augmentation.

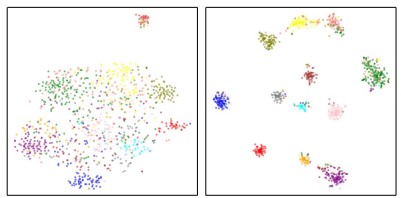

Figure 4: tSNE visualization of the feature space before and after adaptation.

Table 6: Runtime analysis on the VisDA dataset.

| Method | Runtime (s/epoch) | Per-class |
|---|---|---|
| SHOT | 353.53 | 83.0 |
| AaD | 274.36 | 87.3 |
| **SF(DA)**$^2$ (Ours) | 284.49 | 88.1 |
| AaD (5% of $\mathcal{F}, \mathcal{S}$) | 266.76 | 85.9 |
| **SF(DA)**$^2$ (5% of $\mathcal{F}, \mathcal{S}$) | 273.52 | 87.4 |

a degradation in performance. Secondly, applying only FD loss introduces additional constraints to the model by disentangling the feature space. Therefore, the absence of a term utilizing the disentangled space could result in a performance drop.

While the use of a single loss leads to a performance decrease, a synergistic effect arises when IFA loss and FD loss are jointly employed. The disentangled feature space obtained via FD loss allows the estimated covariance matrix to provide directions for class-preserving feature augmentation, and IFA loss performs the feature augmentation. This leads to a notable performance improvement and a further decrease in cosine similarity (compared to SNC+IFA).

We also present the ablation study of the hyperparameters used for the SNC loss in Appendix A.3

**Runtime and tSNE Analysis** We assess our method's efficiency by measuring runtime on the VisDA dataset. The first three rows of Table 6 show our method outperforming SHOT and AaD with less runtime or marginal increase of runtime. The last two rows of Table 6 limit the memory banks $\mathcal{F}$ and $\mathcal{S}$ to 5% of target domain data. Compared to AaD, our method shows more robust performance against the reduction of memory bank size with a similar runtime. Figure 4 visualizes the feature space before and after adaptation on the VisDA dataset, with distinct colors indicating different classes. It clearly shows that target features are effectively clustered after adaptation.

## 5 CONCLUSION

In this work, we propose a novel method for source-free domain adaptation (SFDA), called SF(DA)$^2$, which leverages the intuitions of data augmentation. We propose the spectral neighbor clustering (SNC) loss to find meaningful partitions in the augmentation graph defined on the feature space of the pretrained model. We also propose the implicit feature augmentation (IFA) and feature disentanglement (FD) loss functions to efficiently augment target features in the augmentation graph without linearly increasing computation and memory consumption. Our experiments demonstrate the effectiveness of SF(DA)$^2$ in the SFDA scenario and its superior performance compared to state-of-the-art methods. In the future, we plan to explore the potential of our method for other domain adaptation scenarios and investigate its applicability to other tasks beyond classification.

ACKNOWLEDGMENTS

This work was supported by the National Research Foundation of Korea (NRF) grants funded by the Korea government (Ministry of Science and ICT, MSIT) (2022R1A3B1077720 and 2022R1A5A708390811), Institute of Information & Communications Technology Planning & Evaluation (IITP) grants funded by the Korea government (MSIT) (2021-0-01343: Artificial Intelligence Graduate School Program (Seoul National University) and 2022-0-00959), the BK21 FOUR program of the Education and Research Program for Future ICT Pioneers, Seoul National University in 2024, and 'Regional Innovation Strategy (RIS)' through the National Research Foundation of Korea(NRF) funded by the Ministry of Education(MOE) (2022RIS-005).

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

# A    ADDITIONAL RESULTS

## A.1    ADDITIONAL DATASET

For Office-31, we reproduce SHOT, NRC, CoWA-JMDS, AaD using their official codes. Table 7 presents a comparison between SF(DA)$^2$ and various DA methods, considering both source-present and source-free approaches. Notably, our method exhibits the highest average accuracy among all source-free DA methods.

Table 7: Accuracy (%) on the Office-31 dataset (ResNet-50).

| Method | SF | A→D | A→W | D→A | D→W | W→A | W→D | Avg. |
|---|---|---|---|---|---|---|---|---|
| MCD (Saito et al., 2018) | ✗ | 92.2 | 88.6 | 69.5 | 98.5 | 69.7 | 100.0 | 86.5 |
| CDAN (Long et al., 2018) | ✗ | 92.9 | 94.1 | 71.0 | 98.6 | 69.3 | 100.0 | 87.7 |
| MCC (Jin et al., 2020) | ✗ | 95.6 | 95.4 | 72.6 | 98.6 | 73.9 | 100.0 | 89.4 |
| SRDC (Tang et al., 2020) | ✗ | 95.8 | 95.7 | 76.7 | 99.2 | 77.1 | 100.0 | 90.8 |
| Source only (He et al., 2016) | - | 68.9 | 68.4 | 62.6 | 96.7 | 60.7 | 99.3 | 76.1 |
| SHOT (Liang et al., 2020) | ✓ | 93.8 | 89.6 | 74.5 | 98.9 | 75.3 | 99.9 | 88.7 |
| NRC (Yang et al., 2021a) | ✓ | 92.9 | 93.5 | 76.0 | 98.1 | 75.8 | 99.9 | 89.4 |
| 3C-GAN (Li et al., 2020) | ✓ | 92.7 | 93.7 | 75.3 | 98.5 | 77.8 | 99.8 | 89.6 |
| AaD (Yang et al., 2022) | ✓ | 94.5 | 94.5 | 75.6 | 98.2 | 75.4 | 99.9 | 89.7 |
| **SF(DA)**$^2$ | ✓ | 95.8 | 92.1 | 75.7 | 99.0 | 76.8 | 99.8 | **89.9** |

## A.2    ADDITIONAL METRICS

SFDA methods have used accuracy as the performance metric for evaluation. However, accuracy can sometimes be misleading when used with imbalanced datasets. To mitigate this and provide a more comprehensive evaluation, we add our analysis on the VisDA-RSUT dataset with the harmonic mean of accuracy and the F1-score. In Tables 8 and 9, The results highlight the capability of our method in dealing with imbalanced datasets.

Table 8: Harmonic mean of accuracies (%) on the VisDA-RSUT dataset (ResNet-101).

| Method | Source only | SHOT | CoWA-JMDS | NRC | AaD | **SF(DA)**$^2$ |
|---|---|---|---|---|---|---|
| Harmonic | 16.7 | 45.2 | 61.1 | 61.3 | 65.9 | **70.1** |

Table 9: F1-score (%) on the VisDA-RSUT dataset (ResNet-101).

| Method | plane | bicycle | bus | car | horse | knife | mcycl | person | plant | sktbrd | train | truck | Per-class |
|---|---|---|---|---|---|---|---|---|---|---|---|---|---|
| Source only | 6.2 | 17.0 | 9.8 | 5.3 | 53.6 | 17.2 | 38.1 | 18.9 | 55.3 | 23.7 | 69.3 | 6.3 | 26.7 |
| SHOT | 9.0 | 8.7 | 9.4 | 9.5 | 72.9 | 35.1 | 83.1 | 60.1 | 85.1 | 82.9 | 79.7 | 14.4 | 45.8 |
| CoWA-JMDS | 17.7 | 12.1 | 12.3 | 9.9 | 75.4 | 42.2 | 76.7 | 62.3 | 87.9 | 78.1 | 78.2 | 48.1 | 50.1 |
| NRC | 36.4 | 4.2 | 15.5 | 11.4 | 67.5 | 48.4 | 78.6 | 61.2 | 90.3 | 92.8 | 90.6 | 36.9 | 52.8 |
| AaD | 63.9 | 23.7 | 20.4 | 12.1 | 76.6 | 46.3 | 81.8 | 61.9 | 87.2 | 90.1 | 88.6 | 56.0 | 59.0 |
| **SF(DA)**$^2$ | 77.2 | 53.3 | 12.3 | 13.3 | 79.8 | 49.3 | 85.5 | 67.7 | 91.0 | 90.6 | 87.0 | 49.6 | **63.0** |

## A.3    HYPERPARAMETERS

We present the ablation study of the hyperparameters used in SNC on the VisDA dataset. The second term of the SNC loss ensures diverse predictions for different target features, preventing them from collapsing into one class. During adaptation, weakening the second term can promote a more clustered prediction space and improve performance. Figure 5 (a) shows the ratio of target features that share the same and correct prediction with the 5-nearest neighbors, and Figure 5 (b) shows the average confidence for the predicted labels of the target domain data. As the decaying factor increases from 2 to 5, Figures 5 demonstrate that the target domain data are better clustered in (a) the feature space and (b) the prediction space. Figure 5 (c) shows that our method is robust to the choice of hyperparameters $\beta$ and $K$.

We also present the ablation study of the hyperparameters ($\alpha_1$ and $\alpha_2$) used for IFA and FD losses on the VisDA dataset. As shown in Tables 10 and 11, the model's performance is not sensitive to hyperparameters $\alpha_1$ and $\alpha_2$.

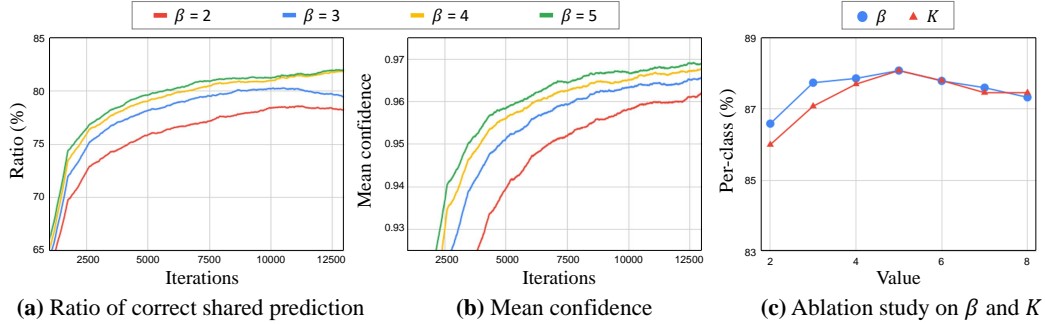

**(a)** Ratio of correct shared prediction **(b)** Mean confidence **(c)** Ablation study on $\beta$ and $K$

Figure 5: Ablation study on hyperparameters for spectral neighbor clustering.

Table 10: Ablation study on the hyperparameter for spectral neighbor clustering.

| $\alpha_1$ | Per-class (%) |
|---|---|
| 1e-3 | 87.3 |
| 1e-4 | 88.1 |
| 1e-5 | 88.0 |

Table 11: Ablation study on the hyperparameter for feature disentanglement.

| $\alpha_2$ | Per-class (%) |
|---|---|
| 1 | 88.0 |
| 10 | 88.1 |
| 20 | 87.4 |

## B  PROOFS

**Proposition 2.** *Suppose that $\tilde{z}_j, \tilde{z}_k \sim \mathcal{N}(z_i, \lambda\Sigma_{\hat{y}_i})$, then we have an upper bound of expected $\mathcal{L}_{\mathrm{EFA}}$ for an infinite number of augmented features, which we call implicit feature augmentation loss:*

$$\mathcal{L}_{\mathrm{EFA}}^{\infty}(\mathbf{z}_i; f, g) = \mathbb{E}_{\tilde{\mathbf{z}}_j \sim \mathcal{N}(\mathbf{z}_i, \lambda\Sigma_{\hat{y}_i})}\left[\mathbb{E}_{\tilde{\mathbf{z}}_k \sim \mathcal{N}(\mathbf{z}_i, \lambda\Sigma_{\hat{y}_i})}\left[-\log \tilde{p}_j^T \tilde{p}_k\right]\right] \tag{11}$$

$$\leq -2\sum_{c=1}^{C} \log \frac{\exp(g(\mathbf{z}_i)_c)}{\sum_{c'=1}^{C} \exp\left(g(\mathbf{z}_i)_{c'} + \frac{\lambda}{2}(w_{c'} - w_c)^T\Sigma_{\hat{y}_i}(w_{c'} - w_c)\right)} = \mathcal{L}_{\mathrm{IFA}}(\mathbf{z}_i, \Sigma_{\hat{y}_i}, g) \tag{12}$$

*where $g(\cdot)_c$ denotes the classifier output (logit) for the c-th class, and $w_c$ is the weight vector for the c-th class of the classifier g.*

*Proof.* Let the classifier $g$ consists of the weight matrix $W = [w_1 \ldots w_C]^T$ and the bias $\mathbf{b} = [b_1 \ldots b_C]$. Now we will derive a closed-form upper bound of expected $\mathcal{L}_{\mathrm{EFA}}$.

$$\mathcal{L}_{\mathrm{EFA}}^{\infty}(\mathbf{z}_i; f, g) = \mathbb{E}_{\tilde{\mathbf{z}}_j \sim \mathcal{N}(\mathbf{z}_i, \lambda\Sigma_{\hat{y}_i})}\left[\mathbb{E}_{\tilde{\mathbf{z}}_k \sim \mathcal{N}(\mathbf{z}_i, \lambda\Sigma_{\hat{y}_i})}\left[-\log \tilde{p}_j^T \tilde{p}_k\right]\right] \tag{13}$$

$$= \mathbb{E}_{\tilde{\mathbf{z}}_j}\left[\mathbb{E}_{\tilde{\mathbf{z}}_k}\left[\sum_{c=1}^{C} -\log \tilde{p}_j^c - \log \frac{\exp(w_c^T\tilde{\mathbf{z}}_k + b_c)}{\sum_{c'=1}^{C}\exp(w_{c'}^T\tilde{\mathbf{z}}_k + b_{c'})}\right]\right] \tag{14}$$

$$= \mathbb{E}_{\tilde{\mathbf{z}}_j}\left[\mathbb{E}_{\tilde{\mathbf{z}}_k}\left[\sum_{c=1}^{C} -\log \tilde{p}_j^c + \log \sum_{c'=1}^{C}\exp\left((w_{c'} - w_c)^T\tilde{\mathbf{z}}_k + (b_{c'} - b_c)\right)\right]\right] \tag{15}$$

$$\leq \mathbb{E}_{\tilde{\mathbf{z}}_j}\left[\sum_{c=1}^{C} -\log \tilde{p}_j^c + \log \sum_{c'=1}^{C}\mathbb{E}_{\tilde{\mathbf{z}}_k}\left[\exp\left((w_{c'} - w_c)^T\tilde{\mathbf{z}}_k + (b_{c'} - b_c)\right)\right]\right] \tag{16}$$

$$= \mathbb{E}_{\tilde{\mathbf{z}}_j}\left[\sum_{c=1}^{C} -\log \tilde{p}_j^c + \log \sum_{c'=1}^{C} e^{(w_{c'}-w_c)^T\mathbf{z}_i + (b_{c'}-b_c) + \frac{\lambda}{2}(w_{c'}-w_c)^T\Sigma_{\hat{y}_i}(w_{c'}-w_c)}\right] \tag{17}$$

$$= \mathbb{E}_{\tilde{\mathbf{z}}_j} \left[ \sum_{c=1}^{C} -\log \tilde{p}_j^c - \log \frac{\exp(w_c^T \mathbf{z}_i + b_c)}{\sum_{c'=1}^{C} \exp \left( w_{c'}^T \mathbf{z}_i + b_{c'} + \frac{\lambda}{2}(w_{c'} - w_c)^T \Sigma_{\hat{y}_i}(w_{c'} - w_c) \right)} \right] \quad (18)$$

$$= \sum_{c=1}^{C} -2 \log \frac{\exp(w_c^T \mathbf{z}_i + b_c)}{\sum_{c'=1}^{C} \exp \left( w_{c'}^T \mathbf{z}_i + b_{c'} + \frac{\lambda}{2}(w_{c'} - w_c)^T \Sigma_{\hat{y}_i}(w_{c'} - w_c) \right)} \quad (19)$$

$$= -2 \sum_{c=1}^{C} \log \frac{\exp(g(\mathbf{z}_i)_c)}{\sum_{c'=1}^{C} \exp \left( g(\mathbf{z}_i)_{c'} + \frac{\lambda}{2}(w_{c'} - w_c)^T \Sigma_{\hat{y}_i}(w_{c'} - w_c) \right)} \quad (20)$$

$$= \mathcal{L}_{\text{IFA}}(\mathbf{z}_i, \Sigma_{\hat{y}_i}, g) \quad (21)$$

where $\tilde{p}_j^c$ denotes the prediction of $\tilde{\mathbf{z}}_j$ for the $c$-th class. Inequality 16 is obtained by applying Jensen's inequality for concave functions (i.e., $\mathbb{E}[\log \sum X] \leq \log \sum \mathbb{E}[X]$) given that the sum of log functions $\sum \log(\cdot)$ is concave. To derive Equation 17, we require the lemma of the moment-generating function for a Gaussian random variable:

**Lemma 2.** *For a Gaussian random variable $X \sim \mathcal{N}(\mu, \sigma^2)$, its moment-generating function is:*

$$\mathbb{E}[e^{tX}] = e^{t\mu + \frac{1}{2}\sigma^2 t^2} \quad (22)$$

In the context of our proof, $(w_{c'} - w_c)^T \tilde{\mathbf{z}}_k + (b_{c'} - b_c)$ is a random variable that follows the Gaussian distribution:

$$(w_{c'} - w_c)^T \tilde{\mathbf{z}}_k + (b_{c'} - b_c) \sim \mathcal{N} \left( (w_{c'} - w_c)^T \mathbf{z}_i + (b_{c'} - b_c), \lambda (w_{c'} - w_c)^T \Sigma_{\hat{y}_i}(w_{c'} - w_c) \right) \quad (23)$$

Lemma 2 then allows us to compute the expectation of this random variable, resulting in:

$$\mathbb{E}_{\tilde{\mathbf{z}}_k} [e^{\left( (w_{c'} - w_c)^T \tilde{\mathbf{z}}_k + (b_{c'} - b_c) \right)}] = e^{(w_{c'} - w_c)^T \mathbf{z}_i + (b_{c'} - b_c) + \frac{\lambda}{2}(w_{c'} - w_c)^T \Sigma_{\hat{y}_i}(w_{c'} - w_c)} \quad (24)$$

Since $\tilde{\mathbf{z}}_j$ and $\tilde{\mathbf{z}}_k$ are drawn from the same distribution, we use the same process to derive Equation 19, concluding the proof.

$\square$

## C  DETAILS

### C.1  IMPLEMENTATION DETAILS

We adopt the backbone of a ResNet-50 for DomainNet, ResNet-101 for VisDA and VisDA-RSUT, and PointNet for PointDA-10. In the final part of the network, we append a fully connected layer, batch normalization (Ioffe & Szegedy, 2015), and the classifier $g$ comprised of a fully connected layer with weight normalization (Salimans & Kingma, 2016). We adopt SGD with momentum 0.9 and train 15 epochs for VisDA, DomainNet, and VisDA-RSUT. We adopt Adam (Kingma & Ba, 2014) and train 100 epochs for PointDA-10. We set batch size to 64 except for DomainNet, where we set it to 128 for a fair comparison. We set the learning rate for VisDA and VisDA-RSUT to 1e-4, 5e-5 for DomainNet, and 1e-6 for PointDA-10, except for the last two layers. Learning rates for the last two layers are increased by a factor of 10, except for PointNet-10 where they are increased by a factor of 2 following NRC (Yang et al., 2021a). Experiments are conducted on a NVIDIA A40.

### C.2  DATASET DETAILS

We use five benchmark datasets for 2D image and 3D point cloud recognition. These include Office-31 (Saenko et al., 2010), with 3 domains (Amazon, Webcam, DSLR), 31 classes, and a total of 15,500 images; VisDA (Peng et al., 2017), with 152K synthetic and 55K real object images across 12 classes; VisDA-RSUT (Li et al., 2021), a subset of VisDA with highly imbalanced classes; DomainNet (Peng et al., 2019), a large-scale benchmark with 6 domains and 345 classes (we select 4 domains (Real, Sketch, Clipart, Painting) with 126 classes, and evaluate SFDA methods on 7 domain shifts following

AdaContrast (Chen et al., 2022).); and PointDA-10 (Qin et al., 2019), a 3D point cloud recognition dataset with 3 domains (namely ModelNet-10, ShapeNet-10, and ScanNet-10), 10 classes, and a total of around 27,700 training and 5,100 test images.

## C.3 ONLINE COVARIANCE ESTIMATION

We follow the method for online covariance estimation proposed in ISDA (Wang et al., 2019) as follows:

$$\boldsymbol{\mu}_c^{(t)} = \frac{n_c^{(t-1)}\boldsymbol{\mu}_c^{(t-1)} + m_c^{(t)}\boldsymbol{\mu}_c'^{(t)}}{n_c^{(t-1)} + m_c^{(t)}} \tag{25}$$

$$\Sigma_c^{(t)} = \frac{n_c^{(t-1)}\Sigma_c^{(t-1)} + m_c^{(t)}\Sigma_c'^{(t)}}{n_c^{(t-1)} + m_c^{(t)}} + \frac{m_c^{(t)} + n_c^{(t-1)}m_c^{(t)}(\boldsymbol{\mu}_c^{(t-1)} - \boldsymbol{\mu}_c'^{(t)})(\boldsymbol{\mu}_c^{(t-1)} - \boldsymbol{\mu}_c'^{(t)})^T}{(n_c^{(t-1)} + m_c^{(t)})^2} \tag{26}$$

$$n_c^{(t)} = n_c^{(t-1)} + m_c^{(t)} \tag{27}$$

where $\boldsymbol{\mu}_c^{(t)}$ and $\Sigma_c^{(t)}$ represent the mean and covariance matrix, respectively, for the $c$-th class at time step $t$. $\boldsymbol{\mu}_c'^{(t)}$ and $\Sigma_c'^{(t)}$ denote the mean and covariance matrix, respectively, of target features which are predicted as the $c$-th class in $t$-th minibatch. $n_c^{(t)}$ is the total number of target features that are predicted as the $c$-th class in all $t$ minibatches, and $m_c^{(t)}$ is the number of target features that are predicted as the $c$-th class in $t$-th minibatch.

# D ADDITIONAL RELATED WORK

## D.1 SPECTRAL CONTRASTIVE LOSS

While we were inspired by the spectral contrastive loss (Spectral CL) (HaoChen et al., 2021) to find partitions of our augmentation graph, there are distinct differences between Spectral CL and our SNC loss.

Spectral CL utilizes augmented data as positive pairs, and data augmentation requires prior knowledge about the dataset. For instance, transforming the color of a lemon to green would turn it into a lime, thus changing the class of the data. Such augmentation that fails to preserve class information can introduce bias to the model, potentially bringing harm to the model's performance (Balestriero et al., 2022). Hence, while Spectral CL requires class-preserving augmentations based on prior knowledge, our SNC loss determines positive pairs without needing explicit augmentation and prior knowledge.

## D.2 IMPLICIT SEMANTIC DATA AUGMENTATION

The IFA loss is motivated by the ISDA (Wang et al., 2019), and we adopted the online estimation of the class-wise covariance matrix from ISDA. However, there are two notable differences between the IFA loss and ISDA.

Firstly, ISDA is designed for supervised learning, where data has labels and no domain shift. In contrast, IFA loss is computed using target domain data without labels and considers SFDA setting with domain shift. To perform implicit feature augmentation under these conditions, IFA loss utilizes pseudo-labels of target domain data which dynamically change during the adaptation process to estimate the class-wise covariance matrix.

Secondly, ISDA derives an upper bound for the expectation of the cross entropy loss for supervised learning. On the other hand, IFA loss is designed to approximate the population augmentation graph, which allows SNC loss to find an improved partition, and it is achieved by deriving an upper bound for the expectation of (the logarithm of) our SNC loss tailored for SFDA.

## D.3 OTHER DOMAIN ADAPTATION METHODS

To address the domain shift problem, domain adaptation has been actively studied, and numerous methods have been proposed. (Yu et al., 2023; Zhao et al., 2020).

NRC (Yang et al., 2021a) and "Connect, not collapse" (CNC) (Shen et al., 2022) are graph-based domain adaptation methods and TSA (Li et al., 2021a) utilizes ISDA for domain adaptation. Our paper presents clear methodological differences and performance advantages compared to existing methods.

NRC (Yang et al., 2021a) is a graph-based source-free method that encourages prediction consistency among neighbors in the feature space by utilizing neighborhood affinity, which is discussed in Section 2. In contrast, our method constructs the augmentation graph in the feature space based on the insights from Intuitions 1 and 2. We then propose SNC loss to identify partitions of this augmentation graph. These methodological differences result in significant performance gaps (e.g., the results on VisDA, PointDA-10, and VisDA-RSUT in Tables 1, 4, and 5).

CNC (Shen et al., 2022) pretrains models using contrastive loss. While CNC depends on data augmentation that needs prior knowledge, our method defines positive pairs without explicit data augmentation. In Table 12, we present a comparative experiment with CNC on 12 domain shifts in the DomainNet dataset (40 classes) and ResNet-50 architecture. The results demonstrate that our approach outperforms the direct application of contrastive loss for pretraining the model in domain adaptation.

Table 12: Accuracy (%) on 12 domain shifts of the DomainNet dataset (40 classes, ResNet-50). For CNC, we brought the performances of SwAV+extra, which was the best-performing contrastive learning method.

| Method | S$\rightarrow$C | S$\rightarrow$P | S$\rightarrow$R | C$\rightarrow$S | C$\rightarrow$P | C$\rightarrow$R | P$\rightarrow$S | P$\rightarrow$C | P$\rightarrow$R | R$\rightarrow$S | R$\rightarrow$C | R$\rightarrow$P | Avg. |
|---|---|---|---|---|---|---|---|---|---|---|---|---|---|
| SwAV+extra (Shen et al., 2022) | 53.5 | 46.8 | 58.1 | 46.2 | 41.7 | 59.4 | 48.7 | 41.3 | 69.0 | 44.6 | 54.2 | 57.3 | 51.7 |
| **SF(DA)$^2$** | 63.8 | 57.2 | 58.6 | 60.3 | 58.9 | 59.8 | 61.4 | 58.0 | 58.2 | 59.5 | 65.2 | 65.4 | **60.4** |

TSA (Li et al., 2021a) employs ISDA for training the source model. Similar to ISDA, it derives an upper bound for the expectation of the cross entropy loss and requires labeled (source domain) data. Conversely, IFA loss is designed to approximate the population augmentation graph, which allows SNC loss to find an improved partition. IFA loss utilizes pseudo-labels of unlabeled target domain data, which dynamically change during the adaptation process, to estimate the class-wise covariance matrix. Using the covariance matrix, IFA loss is derived as a closed-form upper bound for the expectation of (the logarithm of) our SNC loss. The difference in method results in a notable performance gap (e.g., the results in Table 13).

Table 13: Additional performance comparison on the VisDA dataset.

| Method | BSP + TSA | 3C-GAN + HCL | **SF(DA)$^2$** |
|---|---|---|---|
| Per-class | 82.0 | 84.2 | **88.1** |

HCL (Huang et al., 2021) contrasts embeddings from the current model and the historical models. In the paper, 3C-GAN+HCL is the best-performing method on the VisDA dataset, showing a large performance gap compared to our method (e.g., the results in Table 13).

Feat-mixup (Kundu et al., 2022) improves domain adaptation performance via mixup in the feature space with augmented samples. Since this method utilizes data augmentation, it requires prior knowledge for class-preserving image augmentation and multiple forward passes for augmented samples. In contrast, our SNC loss, grounded in Intuitions 1 and 2 in the manuscript, determines positive pairs without explicit augmentation. This difference results in a large performance gap (e.g., the results in Table 3).

Additionally, ProxyMix (Ding et al., 2023) utilizes classifier weights as the class prototypes and proxy features and employs mixup regularization to align the proxy and target domain. PGL (Luo et al., 2023) leverages graph neural networks for open-set domain adaptation.

## E  CODE AVAILABILITY

Code is available in Supplementary Material.

REFERENCES IN APPENDIX

Randall Balestriero, Leon Bottou, and Yann LeCun. The effects of regularization and data augmentation are class dependent. *Advances in Neural Information Processing Systems*, 35:37878–37891, 2022.

Dian Chen, Dequan Wang, Trevor Darrell, and Sayna Ebrahimi. Contrastive test-time adaptation. In *Proceedings of the IEEE/CVF Conference on Computer Vision and Pattern Recognition*, pp. 295–305, 2022.

Yuhe Ding, Lijun Sheng, Jian Liang, Aihua Zheng, and Ran He. Proxymix: Proxy-based mixup training with label refinery for source-free domain adaptation. *Neural Networks*, 167:92–103, 2023.

Jeff Z HaoChen, Colin Wei, Adrien Gaidon, and Tengyu Ma. Provable guarantees for self-supervised deep learning with spectral contrastive loss. *Advances in Neural Information Processing Systems*, 34:5000–5011, 2021.

Kaiming He, Xiangyu Zhang, Shaoqing Ren, and Jian Sun. Deep residual learning for image recognition. In *Proceedings of the IEEE conference on computer vision and pattern recognition*, pp. 770–778, 2016.

Jiaxing Huang, Dayan Guan, Aoran Xiao, and Shijian Lu. Model adaptation: Historical contrastive learning for unsupervised domain adaptation without source data. *Advances in Neural Information Processing Systems*, 34:3635–3649, 2021.

Sergey Ioffe and Christian Szegedy. Batch normalization: Accelerating deep network training by reducing internal covariate shift. In *International conference on machine learning*, pp. 448–456. pmlr, 2015.

Ying Jin, Ximei Wang, Mingsheng Long, and Jianmin Wang. Minimum class confusion for versatile domain adaptation. In *Computer Vision–ECCV 2020: 16th European Conference, Glasgow, UK, August 23–28, 2020, Proceedings, Part XXI 16*, pp. 464–480. Springer, 2020.

Diederik P Kingma and Jimmy Ba. Adam: A method for stochastic optimization. *arXiv preprint arXiv:1412.6980*, 2014.

Jogendra Nath Kundu, Akshay R Kulkarni, Suvaansh Bhambri, Deepesh Mehta, Shreyas Anand Kulkarni, Varun Jampani, and Venkatesh Babu Radhakrishnan. Balancing discriminability and transferability for source-free domain adaptation. In *International Conference on Machine Learning*, pp. 11710–11728. PMLR, 2022.

Rui Li, Qianfen Jiao, Wenming Cao, Hau-San Wong, and Si Wu. Model adaptation: Unsupervised domain adaptation without source data. In *Proceedings of the IEEE/CVF conference on computer vision and pattern recognition*, pp. 9641–9650, 2020.

Shuang Li, Mixue Xie, Kaixiong Gong, Chi Harold Liu, Yulin Wang, and Wei Li. Transferable semantic augmentation for domain adaptation. In *Proceedings of the IEEE/CVF conference on computer vision and pattern recognition*, pp. 11516–11525, 2021a.

Xinhao Li, Jingjing Li, Lei Zhu, Guoqing Wang, and Zi Huang. Imbalanced source-free domain adaptation. In *Proceedings of the 29th ACM International Conference on Multimedia*, pp. 3330–3339, 2021b.

Jian Liang, Dapeng Hu, and Jiashi Feng. Do we really need to access the source data? source hypothesis transfer for unsupervised domain adaptation. In *International Conference on Machine Learning*, pp. 6028–6039. PMLR, 2020.

Jian Liang, Dapeng Hu, Yunbo Wang, Ran He, and Jiashi Feng. Source data-absent unsupervised domain adaptation through hypothesis transfer and labeling transfer. *IEEE Transactions on Pattern Analysis and Machine Intelligence*, 44(11):8602–8617, 2021.

Mingsheng Long, Zhangjie Cao, Jianmin Wang, and Michael I Jordan. Conditional adversarial domain adaptation. *Advances in neural information processing systems*, 31, 2018.

Yadan Luo, Zijian Wang, Zhuoxiao Chen, Zi Huang, and Mahsa Baktashmotlagh. Source-free progressive graph learning for open-set domain adaptation. *IEEE Transactions on Pattern Analysis and Machine Intelligence*, 2023.

Xingchao Peng, Ben Usman, Neela Kaushik, Judy Hoffman, Dequan Wang, and Kate Saenko. Visda: The visual domain adaptation challenge. *arXiv preprint arXiv:1710.06924*, 2017.

Xingchao Peng, Qinxun Bai, Xide Xia, Zijun Huang, Kate Saenko, and Bo Wang. Moment matching for multi-source domain adaptation. In *Proceedings of the IEEE/CVF international conference on computer vision*, pp. 1406–1415, 2019.

Can Qin, Haoxuan You, Lichen Wang, C-C Jay Kuo, and Yun Fu. Pointdan: A multi-scale 3d domain adaption network for point cloud representation. *Advances in Neural Information Processing Systems*, 32, 2019.

Kate Saenko, Brian Kulis, Mario Fritz, and Trevor Darrell. Adapting visual category models to new domains. In *Computer Vision–ECCV 2010: 11th European Conference on Computer Vision, Heraklion, Crete, Greece, September 5-11, 2010, Proceedings, Part IV 11*, pp. 213–226. Springer, 2010.

Kuniaki Saito, Kohei Watanabe, Yoshitaka Ushiku, and Tatsuya Harada. Maximum classifier discrepancy for unsupervised domain adaptation. In *Proceedings of the IEEE conference on computer vision and pattern recognition*, pp. 3723–3732, 2018.

Tim Salimans and Durk P Kingma. Weight normalization: A simple reparameterization to accelerate training of deep neural networks. *Advances in neural information processing systems*, 29, 2016.

Kendrick Shen, Robbie M Jones, Ananya Kumar, Sang Michael Xie, Jeff Z HaoChen, Tengyu Ma, and Percy Liang. Connect, not collapse: Explaining contrastive learning for unsupervised domain adaptation. In *International Conference on Machine Learning*, pp. 19847–19878. PMLR, 2022.

Hui Tang, Ke Chen, and Kui Jia. Unsupervised domain adaptation via structurally regularized deep clustering. In *Proceedings of the IEEE/CVF conference on computer vision and pattern recognition*, pp. 8725–8735, 2020.

Yulin Wang, Xuran Pan, Shiji Song, Hong Zhang, Gao Huang, and Cheng Wu. Implicit semantic data augmentation for deep networks. *Advances in Neural Information Processing Systems*, 32, 2019.

Shiqi Yang, Joost van de Weijer, Luis Herranz, Shangling Jui, et al. Exploiting the intrinsic neighborhood structure for source-free domain adaptation. *Advances in neural information processing systems*, 34:29393–29405, 2021.

Shiqi Yang, Yaxing Wang, Kai Wang, Shangling Jui, et al. Attracting and dispersing: A simple approach for source-free domain adaptation. In *Advances in Neural Information Processing Systems*, 2022.

Zhiqi Yu, Jingjing Li, Zhekai Du, Lei Zhu, and Heng Tao Shen. A comprehensive survey on source-free domain adaptation. *arXiv preprint arXiv:2302.11803*, 2023.

Sicheng Zhao, Xiangyu Yue, Shanghang Zhang, Bo Li, Han Zhao, Bichen Wu, Ravi Krishna, Joseph E Gonzalez, Alberto L Sangiovanni-Vincentelli, Sanjit A Seshia, et al. A review of single-source deep unsupervised visual domain adaptation. *IEEE Transactions on Neural Networks and Learning Systems*, 33(2):473–493, 2020.

