# OpenReview forum: "SF(DA)$^2$: Source-free Domain Adaptation Through the Lens of Data Augmentation"
_ICLR.cc/2024/Conference — ICLR 2024 poster_

### Official Review · Reviewer_jaw9 · 2023-10-22

**Soundness:** 3 good
**Presentation:** 2 fair
**Contribution:** 2 fair
**Rating:** 6
**Confidence:** 4

**Summary:**

This paper addresses source-free domain adaptation task. The submission investigates the problem from the view of data augmentation, along with neighborhood clustering.

**Strengths:**

- The motivation is clear and sound, by using neighborhood (spectral) clustering to address SFDA, which is already proved by the previous related works.

- Instead of utilizing heavy explicit data augmentation, the submission resorts to implicit augmentation via local neighbors. It decreases the complexity.

- Experimental results on several methods are better compared to existing methods.

**Weaknesses:**

- The proposal of SNC is not new. SFDA method AaD already introduced quite similar method (in both motivation and final objective form). More specifically, the proposed SNC just takes [1] into the SFDA task.

- Several results for other methods in the main tables are not identical to the results in their original papers. It is not good to report the reproduced results of own without mentioning it in the paper (though there are always software environment difference).



***Reference***

[1] Provable guarantees for self-supervised deep learning with spectral contrastive loss. NeurIPS 2021

**Questions:**

- The introductions of IFA and FD are the main contribution of the paper. In my understanding, IFA loss plays similar role as the first term in SNC loss, while FD loss plays the similar role as the second term of SNC loss (here FD loss may be better as it quantifies the similarity degrees between different categories). I am just wondering what if only using FD and IFA losses for adaptation.



- In Fig. 2, the results show that the proposed method can not improve SHOT, while in AaD it indicates that SHOT is actually conducting similar operation as NRC and AaD. Do authors have clear explanation about the phenomenon, is it due to the pseudo labeling used in SHOT?

- In Fig. 3, with only FD or IFA on SNC will deteriorate the performance. The authors posit some hypothesis for explanation, while I think there may need more results on several other datasets to prove the assumption. By the way, I think there is no information we could get from Fig. 3(b), since it is neither not consistent with the corresponding accuracy, nor cable to guide the hyperparameter tuning.

- There are two extra hyperparameters for FD and IFA losses, and there seems no ablation study for those hyperparameters. In the paper, it mentioned that $\alpha_1$ and $\alpha_2$ are set to different values for different dataset. I conjecture they may be sensitive on some datasets and hard to tune in the unsupervised way.

---

> ### Author Response · Authors · 2023-11-21
> **Response to Reviewer jaw9 (1/2)**
>
> We thank you for the insightful comments and suggestions. We have addressed each of your questions below.
>
> **Q1.** The proposal of SNC is not new. SFDA method AaD already introduced quite similar method (in both motivation and final objective form). More specifically, the proposed SNC just takes [1, Spectral CL] into the SFDA task.
>
> **Response:** AaD and our method have distinct motivations. We define an augmentation graph based on the given target features. Performing spectral clustering on this augmentation graph identifies meaningful partitions or clusters in the prediction space, which is equivalent to minimizing the SNC loss. While our method and AaD share similar forms for their final loss functions, as you pointed out, our research introduces an innovative perspective on SFDA and significantly expands the scope of research in this area, as acknowledged by Reviewer 44EQ.
>
> Additionally, while we were inspired by the spectral contrastive loss (Spectral CL) to find partitions of our augmentation graph, there are distinct differences between Spectral CL and our SNC loss. Spectral CL utilizes augmented data as positive pairs, and data augmentation requires prior knowledge about the dataset. For instance, transforming the color of a lemon to green would turn it into a lime, thus changing the class of the data. Such augmentation that fails to preserve class information can introduce bias to the model, potentially bringing harm to the model's performance [1]. Hence, while Spectral CL requires class-preserving augmentations based on prior knowledge, our SNC loss determines positive pairs without needing explicit augmentation and prior knowledge.
>
>
> **Q2.** Several results for other methods in the main tables are not identical to the results in their original papers. It is not good to report the reproduced results of own without mentioning it in the paper (though there are always software environment difference).
>
> **Response:** For the VisDA dataset, we reproduced SHOT, NRC, CoWA-JMDS, AaD using their official codes and the pretrained models released by SHOT, and already mentioned it in Section 4.1 of our original manuscript (highlighted in red).
>
> **Q3.** The introductions of IFA and FD are the main contribution of the paper. In my understanding, IFA loss plays similar role as the first term in SNC loss, while FD loss plays the similar role as the second term of SNC loss (here FD loss may be better as it quantifies the similarity degrees between different categories). I am just wondering what if only using FD and IFA losses for adaptation.
>
> **Response:** The IFA loss is derived as the upper bound of the expectation of the first term of the SNC loss. However, the FD loss is not directly related to the second term of the SNC loss. Therefore, we can expect that using only IFA and FD losses may not correctly identify the partition of the augmentation graph, leading to inadequate adaptation. This is confirmed through experiments on the VisDA dataset, which demonstrate an accuracy of 51.4%, in line with our expectations.
>
> **Q4.** In Fig. 2, the results show that the proposed method can not improve SHOT, while in AaD it indicates that SHOT is actually conducting similar operation as NRC and AaD. Do authors have clear explanation about the phenomenon, is it due to the pseudo labeling used in SHOT?
>
> **Response:** Thank you for the good question. IFA and FD losses utilize local neighbor information in the feature space, while NRC and AaD share a similar philosophy. In contrast, SHOT aligns the target features with global source hypotheses for each class. In the AaD paper, the following is mentioned: "In a nutshell, compared to SHOT and BNM, our method first considers the local feature structure to cluster target features." Moreover, IFA and FD losses make use of class-wise covariance matrices, while SHOT does not. This difference in approach between SHOT and IFA can contribute to a decrease in performance.

---

> ### Author Response · Authors · 2023-11-21
> **Response to Reviewer jaw9 (2/2)**
>
> **Q5.** In Fig. 3, with only FD or IFA on SNC will deteriorate the performance. The authors posit some hypothesis for explanation, while I think there may need more results on several other datasets to prove the assumption. By the way, I think there is no information we could get from Fig. 3(b), since it is neither not consistent with the corresponding accuracy, nor cable to guide the hyperparameter tuning.
>
> **Response:** Thanks for your insightful question! IFA identifies directions in the feature space that encodes class semantic information and augments features along those directions. If the feature space is entangled (meaning different directions in the space do not clearly correspond to distinct class semantics), performing feature augmentation in such an entangled feature space may not preserve class information, bringing harm to performance. Applying only FD loss introduces additional constraints to the model by disentangling the feature space. Therefore, the absence of a term utilizing the disentangled space could result in a performance drop.
>
>
> The VisDA-RSUT dataset used for the ablation study has highly imbalanced classes, resulting in an extremely entangled feature space where using only FD or IFA degrades performance. In contrast, in the VisDA dataset with balanced classes, there was no significant performance drop, with results showing SNC: 87.9%, SNC+IFA: 88.0%, and SNC+FD: 87.9%.
>
> In our analysis presented in Section 3.4, we found that the denominator of the IFA loss encourages the kernel space of the model's Jacobian matrix to align with the principal direction of the tangent space of the augmented data manifold, which is in accordance with a finding on data augmentation [2]. Therefore, by comparing SNC (red) and SNC+IFA (yellow) in Fig. 3(b), we can observe a decrease in cosine similarity due to implicit feature augmentation. Moreover, comparing SNC+IFA (yellow) with SNC+IFA+FD (blue) shows that the addition of the FD loss further reduces cosine similarity, indicating that the FD loss facilitates more effective feature augmentation. While SNC+FD (green) might seem redundant for the explanation, it was included to align with Fig. 3(a).
>
> **Q6.** There are two extra hyperparameters for FD and IFA losses, and there seems no ablation study for those hyperparameters. In the paper, it mentioned that a1 and a2 are set to different values for different dataset. I conjecture they may be sensitive on some datasets and hard to tune in the unsupervised way.
>
> **Response:** Thanks for the suggestion! To report the optimal performance for each dataset, we conducted slight hyperparameter tuning. However, in six datasets, we experimented with only two settings for each hyperparameter ($\alpha_1$=1e-4, 1e-3) ($\alpha_2$=1, 10), achieving state-of-the-art performance. The model's performance is not sensitive to hyperparameters $\alpha_1$ and $\alpha_2$. As you suggested, we newly conducted an ablation study on $\alpha_1$ and $\alpha_2$, and the results on the VisDA dataset are presented in the following tables:
>
> <Table S5.> Ablation study on $\alpha_1$
>
> |$\alpha_1$|Per-class (%)|
> |:---:|:---:|
> |1e-3|87.3|
> |1e-4|88.1|
> |1e-5|88.0|
>
> <Table S6.>  Ablation study on $\alpha_2$
>
> |$\alpha_2$|Per-class (%)|
> |:---:|:---:|
> |1|88.0|
> |10|88.1|
> |20|87.4|
>
> We added the additional ablation studies in Tables 10 and 11 in the revision.
>
> We hope that our response has addressed your concerns. Thank you very much!
>
> [1] The effects of regularization and data augmentation are class dependent, NeurIPS 2022.
>
> [2] A Data-Augmentation Is Worth A Thousand Samples: Analytical Moments And Sampling-Free Training, NeurIPS 2022.

---

> ### Comment · Reviewer_jaw9 · 2023-11-22
>
> Thanks for the effort to address my concerns. I improve the score from 5 to 6.

---

> > ### Author Response · Authors · 2023-11-23
> > **Thank you!**
> >
> > We appreciate your response. We are sincerely glad to learn that you have been satisfied with the additional experiments and discussion.

---

### Official Review · Reviewer_s4wJ · 2023-10-26

**Soundness:** 3 good
**Presentation:** 3 good
**Contribution:** 2 fair
**Rating:** 5
**Confidence:** 4

**Summary:**

This paper proposes a novel approach, namely Source-Free Domain Adaptation through the lens of Data Augmentation (SF(DA)^2), to address the challenge of source-free domain adaptation. The proposed method aims to harness the advantages of data augmentation while mitigating the drawbacks associated with relying on prior knowledge of class-preserving transformations and the increased memory and computational demands. SF(DA)^2 comprises three key components: Spectral Neighborhood Clustering (SNC) loss, Implicit Feature Augmentation (IFA) loss, and Features Disentanglement (FD) loss. The SNC loss is designed to cluster the target data in the prediction space, the IFA loss emulates the effects of augmented features without imposing additional computation and memory overhead, and the FD loss captures distinct class semantic information along diverse directions. Rigorous experimentation on various benchmarks verifies that the proposed method attains state-of-the-art performance in source-free domain adaptation.

**Strengths:**

(1) The writing of this paper is good.

(2) The method proposed is straightforward yet effective, and the results obtained are at the forefront of current research in the field.

**Weaknesses:**

(1)	What are the differences between the proposed SNC loss and the loss function in the existing AaD [1]? As far as I know, the purpose of the loss in AaD is to encourage similar neighbors to be close to each other and dissimilar neighbors to be far apart within a batch. This is similar to the SNC loss proposed in this paper. The author needs to clarify the differences between these two loss functions.

(2)	Are the estimated covariance matrices accurate in the IFA and FD losses, given the domain shift between the source and target domains? The estimation of these matrices relies on the output probabilities of the target data. However, this estimation process may introduce biases, especially when there are significant dissimilarities between the classes in the target domain and those in the source domain. As a result, the probabilities associated with these classes in the target domain, as generated by the source domain model, may be greatly reduced. Consequently, there is a potential risk of losing the capability to effectively learn these specific classes during the subsequent learning process, presenting a challenge to rectify the situation.

(3)	IFA operations have shown promise as a universally applicable data augmentation technique in unsupervised tasks. Can IFA be applied to other existing unsupervised tasks? Can you provide examples to validate its effectiveness?

(4)	Two questions regarding the experiment:
(a) In VisDA, Domain-Net, and PointDA-10 datasets, IFA loss has a weight of 1e-4, while FD loss has a weight of 10, resulting in a limited impact of IFA loss on the training process compared to FD loss. Can the inclusion of the other two datasets, apart from the VisDA-Rust task, demonstrate the significance of this new data augmentation technique?
(b) Why are there no comparisons of NRC and AaD methods on the DomainNet dataset? Similarly, why are the results of AaD not compared on the PointDA-10 dataset?

(5)	In the Introduction, the authors mention that recent studies have used data augmentation techniques to improve adaptation performance. However, there is a lack of more relevant work cited, such as [2] and [3].

[1] Yang, S., Jui, S., & van de Weijer, J. (2022). Attracting and dispersing: A simple approach for source-free domain adaptation. Advances in Neural Information Processing Systems, 35, 5802-5815.

[2] Ding, Y., Sheng, L., Liang, J., Zheng, A., & He, R. (2023). Proxymix: Proxy-based mixup training with label refinery for source-free domain adaptation. Neural Networks, 167, 92-103.

[3] Luo, Y., Wang, Z., Chen, Z., Huang, Z., & Baktashmotlagh, M. (2023). Source-free progressive graph learning for open-set domain adaptation. IEEE Transactions on Pattern Analysis and Machine Intelligence.

**Questions:**

Please see the weaknesses.

---

> ### Author Response · Authors · 2023-11-21
> **Response to Reviewer s4wJ (1/2)**
>
> We thank you for the insightful comments and suggestions. We have addressed each of your questions below.
>
> **Q1.** What are the differences between the proposed SNC loss and the loss function in the existing AaD [1]? As far as I know, the purpose of the loss in AaD is to encourage similar neighbors to be close to each other and dissimilar neighbors to be far apart within a batch. This is similar to the SNC loss proposed in this paper. The author needs to clarify the differences between these two loss functions.
>
> **Response:** AaD and our method have distinct motivations. We define an augmentation graph based on the given target features. Performing spectral clustering on this augmentation graph identifies meaningful partitions or clusters in the prediction space, which is equivalent to minimizing the SNC loss. While our method and AaD share similar forms for their final loss functions, as you pointed out, our research introduces an innovative perspective on SFDA and significantly expands the scope of research in this area, as acknowledged by Reviewer 44EQ.
>
> **Q2.** Are the estimated covariance matrices accurate in the IFA and FD losses, given the domain shift between the source and target domains? The estimation of these matrices relies on the output probabilities of the target data. However, this estimation process may introduce biases, especially when there are significant dissimilarities between the classes in the target domain and those in the source domain. As a result, the probabilities associated with these classes in the target domain, as generated by the source domain model, may be greatly reduced. Consequently, there is a potential risk of losing the capability to effectively learn these specific classes during the subsequent learning process, presenting a challenge to rectify the situation.
>
> **Response:** Thank you for the good question. The IFA loss estimates the covariance matrix using the given target features and pseudo-labels for implicit feature augmentation. However, during the initial adaptation phase, the feature space tends to become entangled due to domain shift, which may lead to an inaccurate covariance matrix and potentially degrade model performance. Therefore, we designed the influence of the estimated covariance matrix to be small during early training and gradually increase as training progresses (as indicated by \lambda in Equation 5). Additionally, to further disentangle the feature space, we introduced the FD loss and regularized the features to distribute in different directions for each class.
>
>  If a pretrained model has been trained well, each direction in the feature space is likely to contain valuable class semantic information, and one might expect the cosine distances between groups of features belonging to the same ground-truth class to be close. Even in cases where domain shift may lead to incorrect assignments, the FD loss can be expected to reduce the cosine distance between groups of features from the same class during the process of disentangling the feature space.
>
> However, for two groups of features belonging to the same ground-truth category to be assigned to different partitions, one group of features must possess features that are more similar to those of other classes than to its ground-truth class. In such scenarios, it is challenging to achieve unsupervised domain adaptation without supervision.
>
> **Q3.** IFA operations have shown promise as a universally applicable data augmentation technique in unsupervised tasks. Can IFA be applied to other existing unsupervised tasks? Can you provide examples to validate its effectiveness?
>
> **Response:** We appreciate your suggestion. The IFA loss is derived from an augmentation graph in the feature space, taking into account a pretrained model and domain shift. Applying this to different unsupervised tasks would require a redefinition of the augmentation graph and the derivation of a new loss tailored to the specific task. Therefore, we consider this to fall beyond the scope of this paper. Nonetheless, we appreciate your suggestion and believe it could be an interesting avenue for future work.

---

> ### Author Response · Authors · 2023-11-21
> **Response to Reviewer s4wJ (2/2)**
>
> **Q4.** In VisDA, Domain-Net, and PointDA-10 datasets, IFA loss has a weight of 1e-4, while FD loss has a weight of 10, resulting in a limited impact of IFA loss on the training process compared to FD loss. Can the inclusion of the other two datasets, apart from the VisDA-Rust task, demonstrate the significance of this new data augmentation technique?
>
> **Response:** Since the two losses serve as different types of regularizers, it's difficult to claim that IFA's impact is small solely due to the difference in weight. (For instance, the hyperparameter for weight decay commonly used in neural networks typically has a scale around 1e-3, whereas the hyperparameter for gradient penalty used in GANs often uses a scale of about 10. However, employing both doesn't necessarily mean that the impact of the gradient penalty is substantially stronger.) As depicted in Figure 3(a), even adding just 1e-3 of IFA results in a performance increase of over 5%p. While consistent performance improvements were observed in other datasets (from 87.9% to 88.1% in VisDA and from 68.0% to 68.3% in DomainNet), the enhancement is particularly significant in the highly imbalanced dataset. This notable performance boost in imbalanced datasets is believed to be due to the FD loss enabling distinct covariance matrices for each class and utilizing this for feature augmentation, thereby allowing minority classes to learn in distinct directions within the feature space.
>
> **Q5.** Why are there no comparisons of NRC and AaD methods on the DomainNet dataset? Similarly, why are the results of AaD not compared on the PointDA-10 dataset?
>
> **Response:** We appreciate your suggestion. Following your suggestion, we newly compared our method with NRC and AaD on the DomainNet and PointDA-10 datasets and the results are presented in the following tables:
>
> <Table S3.> Accuracy on the DomainNet dataset.
>
> |Method|S$\rightarrow$P|C$\rightarrow$S|P$\rightarrow$C|P$\rightarrow$R|R$\rightarrow$S|R$\rightarrow$C|R$\rightarrow$P|Avg.|
> |:---:|:---:|:---:|:---:|:---:|:---:|:---:|:---:|:---:|
> |NRC|65.7|58.6|64.5|82.3|58.4|65.2|68.2|66.1|
> |AaD|65.4|54.2|59.8|81.8|54.6|60.3|68.5|63.5|
> |**SF(DA)$^2$**|67.7|59.6|67.8|83.5|60.2|68.8|70.5|**68.3**|
>
> <Table S4.> Accuracy on the PointDA-10 dataset.
>
> |Method|Model$\rightarrow$Shape|Model$\rightarrow$Scan|Shape$\rightarrow$Model|Shape$\rightarrow$Scan|Scan$\rightarrow$Model|Scan$\rightarrow$Shape|Avg.|
> |:---:|:---:|:---:|:---:|:---:|:---:|:---:|:---:|
> |AaD|69.6|34.6|67.7|28.8|68.0|66.6|55.9|
> |**SF(DA)$^2$**|70.3|35.5|68.3|29.0|70.4|69.2|**57.1**|
>
> We added the additional ablation studies in Tables 2 and 4 in the revision.
>
> **Q6.** In the Introduction, the authors mention that recent studies have used data augmentation techniques to improve adaptation performance. However, there is a lack of more relevant work cited, such as [2] and [3].
>
> **Response:** Thank you for suggesting additional related work. In the revision, we included these related works in Appendix D.3 (highlighted in red).
>
> We hope that our response has addressed your concerns. Thank you very much!

---

> ### Author Response · Authors · 2023-11-23
> **Looking forward to your post-rebuttal feedback!**
>
> Dear Reviewer s4wJ,
>
> Thank you again for the insightful comments and suggestions! Given the limited time remaining, we eagerly anticipate your subsequent feedback. It would be our pleasure to offer more responses to further demonstrate the effectiveness of our methodology.
>
> In our previous response, we have thoroughly reviewed your comments and provided responses summarized as follows:
>
> - Explained the differences between SNC loss and AaD loss. SNC loss involves spectral clustering on the augmentation graph based on target features, offering a new perspective on SFDA.
> - Explained how our method mitigates bias in the estimated covariance matrices, enhancing accuracy in domain adaptation.
> - Conducted the new ablation study on the IFA loss and highlighted the notable performance improvement in the imbalanced dataset, facilitated by learning distinct feature directions.
> - Conducted the new comparisons of NRC and AaD methods on the DomainNet and PointDA-10 datasets, showing the superior performance of our method.
> - Included more related works on data augmentation techniques for domain adaptation.
>
> We hope that the provided new experiments and the additional explanation have convinced you of the merits of this paper. If there are additional questions, please feel free to let us know.
>
> Additionally, we wish to express our gratitude once again to you for your insightful feedback. Incorporating your suggestions has undoubtedly enhanced the clarity and robustness of our work.
>
> We deeply appreciate your time and effort!
>
> Best regards, Authors

---

> ### Author Response · Authors · 2023-11-23
> **A kind reminder**
>
> Dear reviewer s4wJ
>
> The interactive discussion phase will end in few hours, and we cannot have discussions with you anymore after the deadline. We wish that our response has addressed your concerns, and turns your assessment to a more positive side. Please let us know if there are any other things that we need to clarify.
>
> We thank you so much for your helpful and insightful suggestion.
>
> Best, Authors

---

> ### Comment · Reviewer_s4wJ · 2023-11-23
>
> Thanks for the effort to address my concerns. I would like to improve the score.

---

> > ### Author Response · Authors · 2023-11-23
> > **Thank you!**
> >
> > We deeply appreciate your response. We are sincerely glad to learn that you have been satisfied with the additional experiments and discussion.

---

### Official Review · Reviewer_44EQ · 2023-10-29

**Soundness:** 3 good
**Presentation:** 3 good
**Contribution:** 3 good
**Rating:** 8
**Confidence:** 4

**Summary:**

This work proposed a novel perspective of solving source-free domain adaptation (SFDA) problems through implicit feature augmentation on augmentation graphs. Motivated by the two assumptions, the authors naturally constructed an augmentation graph within the feature space. Initially, the augmentation graph is formed based on neighboring features. Subsequently, it is harnessed to identify clusters within the feature space using the SNC loss. With high-quality clusters, target features are implicitly augmented by an EFA loss as an upper bound of the first term of SNC loss instead of directly sampling from Gaussian distribution. Considering similar categories, feature space is further disentangled by maximizing cosine distances, leading to preserved class semantics. Experiments and analysis were conducted to prove the effectiveness of the proposed method.

**Strengths:**

Originality: This submission presents an innovative perspective on SFDA, introducing implicit augmentation without the need for prior knowledge. This novel approach has been thoughtfully considered and significantly broadens the scope of research in this area.

Quality: The proposed method's quality is supported by a comprehensive body of theoretical and experimental evidence.

Clarity: The definitions and mathematical terms are highly connected with the corresponding elements of the proposed method. The presentation and logical flow are well-executed.

Significance: The application of data augmentation to SFDA from a novel standpoint is of great significance. While the theoretical contributions to the SFDA field may be limited, the method holds practical and theoretical value, as it is built upon sound mathematical foundations and addresses practical issues, such as handling similar categories.

**Weaknesses:**

A more detailed description of the relationship between the augmentation graph and section 3.4 is warranted, as there appears to be no explicit utilization of G_hat in section 3.4.

How does equation (4) influence G_hat, and what is the impact of equation (4) and G_hat on the overall optimization process, given that equation (4) only considers cosine-similarity neighborhoods and mini-batch statistics?

Within spectral clustering methods, could the scenario wherein two groups of features belonging to the same ground truth category are erroneously assigned to separate partitions ultimately lead to a degradation in model performance, as they may be pushed apart within the feature space?

It would be much appreciated if some DA surveys and reviews are included to better show the background and related work, such as "A Comprehensive Survey on Source-free Domain Adaptation", "A Review of Single-Source Deep Unsupervised Visual Domain Adaptation".

The compared baselines are obviously insufficient. For source-free DA, some typical and latest methods are not compared, such as SHOT++. For source-available DA, the compared baselines are too old. Some more baselines published in 2022 and 2023 are required but missing.

**Questions:**

Please see the weaknesses above.

---

> ### Author Response · Authors · 2023-11-21
> **Response to Reviewer 44EQ**
>
> We thank you for the insightful comments and suggestions. We have addressed each of your questions below.
>
> **Q1.** A more detailed description of the relationship between the augmentation graph and section 3.4 is warranted, as there appears to be no explicit utilization of G_hat in section 3.4.
>
>  **Response:** G_hat is an instance of the augmentation graph defined from the given target features, and obtaining the partition of G_hat on the prediction space leads to the derivation of the SNC loss. Implicit feature augmentation (section 3.4) goes beyond the given target features to approximate the population augmentation graph (G), and derive the IFA loss by obtaining a closed-form upper bound for the first term of the expected SNC loss in a tractable way from G. In other words, G_hat is defined from a finite number of given target features, while IFA has the effect of augmenting an infinite number of target features.
>
> **Q2.** How does equation (4) influence G_hat, and what is the impact of equation (4) and G_hat on the overall optimization process, given that equation (4) only considers cosine-similarity neighborhoods and mini-batch statistics?
>
>  **Response:** Thank you for the good question! G_hat is an instance of the augmentation graph defined using cosine distances between the given target features. Performing spectral clustering on this augmentation graph finds meaningful partitions or clusters in the prediction space, which is equivalent to minimizing the SNC loss (Equation 4). In other words, minimizing the SNC loss allows us to discover meaningful clusters of target features in the prediction space, and an optimal solution would result in one-hot vectors for each class. Therefore, successfully leveraging the relationships between target features enables effective source-free domain adaptation.
>
> **Q3.** Within spectral clustering methods, could the scenario wherein two groups of features belonging to the same ground truth category are erroneously assigned to separate partitions ultimately lead to a degradation in model performance, as they may be pushed apart within the feature space?
>
> **Response:** If a pretrained model has been trained well, each direction in the feature space is likely to contain valuable class semantic information, and one might expect the cosine distances between groups of features belonging to the same ground-truth class to be close. Even in cases where domain shift may lead to incorrect assignments, the FD loss can be expected to reduce the cosine distance between groups of features from the same class during the process of disentangling the feature space.
>
> However, for two groups of features belonging to the same ground-truth category to be assigned to different partitions, one group of features must possess features that are more similar to those of other classes than to its ground-truth class. In such scenarios, it is challenging to achieve unsupervised domain adaptation without supervision.
>
> **Q4.** It would be much appreciated if some DA surveys and reviews are included to better show the background and related work, such as "A Comprehensive Survey on Source-free Domain Adaptation", "A Review of Single-Source Deep Unsupervised Visual Domain Adaptation".
>
> **Response:** Thank you for suggesting additional related work. In the revision, we included these related works in Appendix D.3 (highlighted in red).
>
> **Q5.** The compared baselines are obviously insufficient. For source-free DA, some typical and latest methods are not compared, such as SHOT++. For source-available DA, the compared baselines are too old. Some more baselines published in 2022 and 2023 are required but missing.
>
> **Response:** We already compared our method with SHOT++ in Table 3 of our manuscript and our method outperforms SHOT++ by a large margin (2.3%p). We also compared our method with the latest methods up to 2022, both in the main text and the appendix. Additionally, we newly compared our method with NRC and AaD on the DomainNet and PointDA-10 datasets and the results are presented in the following tables:
>
> <Table S1.> Accuracy on the DomainNet dataset.
>
> |Method|S$\rightarrow$P|C$\rightarrow$S|P$\rightarrow$C|P$\rightarrow$R|R$\rightarrow$S|R$\rightarrow$C|R$\rightarrow$P|Avg.|
> |:---:|:---:|:---:|:---:|:---:|:---:|:---:|:---:|:---:|
> |NRC|65.7|58.6|64.5|82.3|58.4|65.2|68.2|66.1|
> |AaD|65.4|54.2|59.8|81.8|54.6|60.3|68.5|63.5|
> |**SF(DA)$^2$**|67.7|59.6|67.8|83.5|60.2|68.8|70.5|**68.3**|
>
> <Table S2.> Accuracy on the PointDA-10 dataset.
>
> |Method|Model$\rightarrow$Shape|Model$\rightarrow$Scan|Shape$\rightarrow$Model|Shape$\rightarrow$Scan|Scan$\rightarrow$Model|Scan$\rightarrow$Shape|Avg.|
> |:---:|:---:|:---:|:---:|:---:|:---:|:---:|:---:|
> |AaD|69.6|34.6|67.7|28.8|68.0|66.6|55.9|
> |**SF(DA)$^2$**|70.3|35.5|68.3|29.0|70.4|69.2|**57.1**|
>
> We added the additional ablation studies in Tables 2 and 4 in the revision.
>
>
> We hope that our response has addressed your concerns. Thank you very much!

---

> > ### Comment · Reviewer_44EQ · 2023-11-23
> > **after rebuttal**
> >
> > Thank the authors for providing the rebuttal to answer my questions. My concerns have been addressed and I would like to increase my rating.

---

> > > ### Author Response · Authors · 2023-11-23
> > > **Thank you!**
> > >
> > > Thank you very much for your response. We are glad to know that your concerns have been addressed.

---

> ### Author Response · Authors · 2023-11-23
> **Looking forward to your post-rebuttal feedback!**
>
> Dear Reviewer 44EQ,
>
> Thank you again for the insightful comments and suggestions! Given the limited time remaining, we eagerly anticipate your subsequent feedback. It would be our pleasure to offer more responses to further demonstrate the effectiveness of our methodology.
>
> In our previous response, we have thoroughly reviewed your comments and provided responses summarized as follows:
>
> - Explained the relationship between the augmentation graph and implicit feature augmentation.
> - Explained the relationship between the augmentation graph and SNC loss, and their impact on the overall optimization.
> - Explained how our method mitigates incorrect partition assignments, enhancing accuracy in domain adaptation.
> - Included the suggested review and survey on domain adaptation.
> - Conducted the new comparisons of recent methods (NRC and AaD) on the DomainNet and PointDA-10 datasets, showing the superior performance of our method.
>
> We hope that the provided new experiments and the additional explanation have convinced you of the merits of this paper. If there are additional questions, please feel free to let us know.
>
> Additionally, we wish to express our gratitude once again to you for your insightful feedback. Incorporating your suggestions has undoubtedly enhanced the clarity and robustness of our work.
>
> We deeply appreciate your time and effort!
>
> Best regards, Authors

---

### Official Review · Reviewer_7eAE · 2023-11-01

**Soundness:** 3 good
**Presentation:** 3 good
**Contribution:** 3 good
**Rating:** 6
**Confidence:** 4

**Summary:**

This work presents a novel approach to address the source-free domain adaptation task, which aims to adapt the pre-trained model to suit the unlabelled target domain with distribution shifts. The work looks at a novel perspective – from data augmentation in latent feature space instead of applying transformation of raw data to save computational costs while improving model robustness. The SNC loss is proposed to learn discriminative features and form tighter clusters; To augment target features, a class-wise covariance matrix is estimated in an online manner with the help of pseudo labels. IFA loss promotes the decision boundaries for each class to align with the principal direction of the variance of target features. To encourage each direction to be orthogonal, FD loss is further added. Comprehensive experiments are provided on both 2D and 3D datasets and evidence the validity of the proposed approach.

---- Post Rebuttal ----
Thanks for the detailed explanation the authors provided. I would like to maintain the original positive rating for this work.

**Strengths:**

+ The paper looks at an interesting way of exploring augmentation in SFDA, where the feature diversity is guaranteed while the resulting computational costs are not significantly increased compared to conventional augmentation methods. The main idea is straightforward and effective, and the authors also bring in-depth discussions and theoretical analysis to provide insights into why the idea works. This makes the proposed well-motivated and inspiring.
+ The paper presents a large number of experimental comparisons on different benchmark datasets and demonstrates a significant boost over previous methods. The running time analysis gives clear support to what is claimed in the introduction section.
+ The paper is well structured and notations are used properly.

**Weaknesses:**

- As shown in Figure 3, the ablation study reveals that solely optimizing SNC performs better than with either FD or IFA loss. It remains unclear why SNC+IFA is inferior as augmented views are provided. More discussions can be shared to better help readers understand the proposed approach.

**Questions:**

My question mainly lies in (1) the ablation study: why do two regularizations need to be applied simultaneously? (2) the confusion matrix is estimated in an online way. How to guarantee its quality/accuracy?

If the questions are properly addressed, I will consider raising my rating.

---

> ### Author Response · Authors · 2023-11-21
> **Response to Reviewer 7eAE**
>
> We thank you for the insightful comments and suggestions. We have addressed each of your questions below.
>
> **Q1.** The ablation study: why do two regularizations need to be applied simultaneously?
>
> **Response:** Thanks for your insightful question! Data augmentation that fails to preserve class information can introduce bias to the model. For instance, transforming the color of a lemon to green would turn it into a lime, potentially bringing harm to the model's performance [1].
>
>
> IFA identifies directions in the feature space that encodes class semantic information and augments features along those directions. If the feature space is entangled (meaning different directions in the space do not clearly correspond to distinct class semantics), performing feature augmentation in such an entangled feature space may not preserve class information, bringing harm to performance. Applying only FD loss introduces additional constraints to the model by disentangling the feature space. Therefore, the absence of a term utilizing the disentangled space could result in a performance drop.
>
>
> However, a synergistic effect arises when IFA loss and FD loss are jointly employed. The disentangled feature space obtained via FD loss allows the estimated covariance matrix to provide directions for class-preserving feature augmentation. Subsequently, the IFA loss performs the feature augmentation, leading to a notable performance improvement.
>
> **Q2.** the confusion matrix is estimated in an online way. How to guarantee its quality/accuracy?
>
> **Response:** The IFA loss estimates the covariance matrix using the given target features and pseudo-labels for implicit feature augmentation. However, during the initial adaptation phase, the feature space tends to become entangled due to domain shift, which may lead to an inaccurate covariance matrix and potentially degrade model performance. Therefore, we designed the influence of the estimated covariance matrix to be small during early training and gradually increase as training progresses (as indicated by \lambda in Equation 5). Additionally, to further disentangle the feature space, we introduced the FD loss and regularized the features to distribute in different directions for each class.
>
> We hope that our response has addressed your concerns. Thank you very much!
>
> [1] The Effects of Regularization and Data Augmentation are Class Dependent, NeurIPS 2022.

---

> ### Author Response · Authors · 2023-11-23
> **Looking forward to your post-rebuttal feedback!**
>
> Dear Reviewer 7eAE,
>
> Thank you again for the insightful comments and suggestions! Given the limited time remaining, we eagerly anticipate your subsequent feedback. It would be our pleasure to offer more responses to further demonstrate the effectiveness of our methodology.
>
> In our previous response, we have thoroughly reviewed your comments and provided responses summarized as follows:
>
> - Explained the necessity of simultaneous application of two regularizers (IFA and FD losses).
> - Explained how our method ensures the quality of the estimated covariance matrix, enhancing accuracy in domain adaptation.
>
> We hope that the provided new experiments and the additional explanation have convinced you of the merits of this paper. If there are additional questions, please feel free to let us know.
>
> Additionally, we wish to express our gratitude once again to you for your insightful feedback. Incorporating your suggestions has undoubtedly enhanced the clarity and robustness of our work.
>
> We deeply appreciate your time and effort!
>
> Best regards, Authors

---

> > ### Author Response · Authors · 2023-11-23
> > **A kind reminder**
> >
> > Dear reviewer 7eAE
> >
> > The interactive discussion phase will end in few hours, and we cannot have discussions with you anymore after the deadline. We wish that our response has addressed your concerns, and turns your assessment to a more positive side. Please let us know if there are any other things that we need to clarify.
> >
> > We thank you so much for your helpful and insightful suggestion.
> >
> > Best, Authors

---

### Meta-Review · Area_Chair_dnu1 · 2023-12-06

**Metareview:**

This study introduces a novel approach to address source-free domain adaptation (SFDA) challenges by leveraging implicit feature augmentation on augmentation graphs. The authors construct an augmentation graph within the feature space based on neighboring features and employ the SNC loss to identify high-quality clusters. Target features are then implicitly augmented using an EFA loss, enhancing the effectiveness of the proposed method. Experimental results demonstrate its efficacy, although some questions remain regarding the performance of SNC+IFA compared to augmented views. Despite these concerns, the work contributes to the SFDA field by presenting a challenging and realistic setting with potential interest to the research community.

**Justification For Why Not Higher Score:**

This work provides an interesting view to solve the source-free domain adaptation, it has some insights. However, it is not enough to be selected as spolight.

**Justification For Why Not Lower Score:**

This work  looks at an interesting way of exploring augmentation in SFDA, it fits the requirement of ICLR. Absolutely, it can provide some insights to the research community. So it should be accepted .

---

### Decision · Program_Chairs · 2024-01-16

Accept (poster)